# Unifying Symbolic Music Arrangement: Track-Aware Reconstruction and Structured Tokenization

**Longshen Ou**[♯]  **Jingwei Zhao**[♯]  **Ziyu Wang**[♭,♮]  **Gus Xia**[♭]
**Qihao Liang**[♯]  **Torin Hopkins**[♮]  **Ye Wang**[♯]

[♯]Sound and Music Computing Lab, School of Computing, NUS
[♭]Courant Institute of Mathematical Sciences, New York University
[♮]Music X Lab, MBZUAI

## Abstract

We present a unified framework for automatic multitrack music arrangement that enables a single pre-trained symbolic music model to handle diverse arrangement scenarios, including reinterpretation, simplification, and additive generation. At its core is a segment-level reconstruction objective operating on token-level disentangled content and style, allowing for flexible any-to-any instrumentation transformations at inference time. To support track-wise modeling, we introduce *REMI-z*, a structured tokenization scheme for multitrack symbolic music that enhances modeling efficiency and effectiveness for both arrangement tasks and unconditional generation. Our method outperforms task-specific state-of-the-art models on representative tasks in different arrangement scenarios—band arrangement, piano reduction, and drum arrangement, in both objective metrics and perceptual evaluations. Taken together, our framework demonstrates strong generality and suggests broader applicability in symbolic music-to-music transformation.[1]

## 1 Introduction

Music arrangement is the art of adapting compositions for performance contexts that differ from their original forms [4]. It plays a central role in many music creation process, including professional production, live performance, music education, and digital content creation. Automating this process can expand music accessibility and accelerate music creation. Although arrangement forms vary—e.g., rewriting for different instruments (reinterpretation) [9], simplifying for solo performance (reduction) [29, 30], or adding new tracks (additive generation) [20, 19]—they share a common structure: generating new music tracks conditioned on existing ones under explicit content and instrument constraints. However, prior work typically addresses each task independently [42, 30, 29, 19], using specialized model architectures and training schemes. Such designs lack cross-task generality, increase implementation cost, and fail to leverage the musical knowledge learned by large-scale pre-trained symbolic models that could potentially improve the arrangement quality.

Meanwhile, generative modeling of symbolic music, i.e., music in notation-based formats, has advanced rapidly with large-scale pre-trained models that capture rich musical styles and structures via autoregressive modeling [17, 8, 23, 37, 32, 22]. Inspired by natural language processing, these models scale to billions of parameters and are trained on vast corpora. Although their unconditional generation quality has improved substantially, their applications in real-world conditional generation remains relatively limited. Prior work primarily targets coarse-level control, such as style [3, 17], structure [40], polyphony level [22], or sentiment [18], while fine-grained conditioning on existing musical content—precisely what arrangement tasks demand—remains underexplored.

---

[1]Demos and code: `https://www.oulongshen.xyz/automatic_arrangement`.

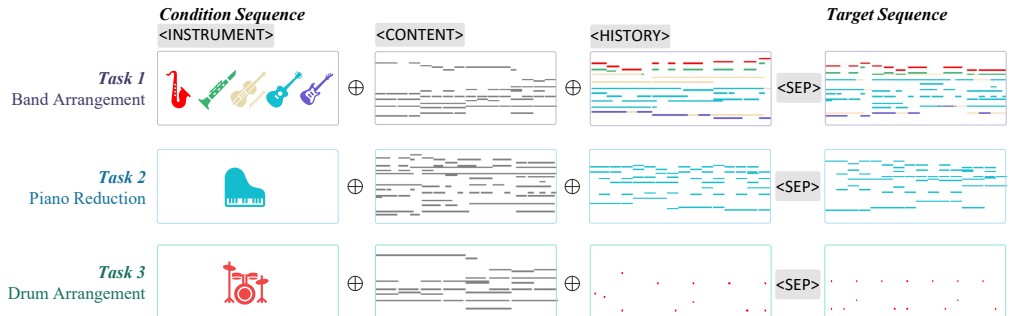

Figure 1: Overview of the proposed unified framework for each arrangement task. The symbol ⊕ denotes concatenation of component sequences. Music segments are decomposed into three subsequences: *instruments*, *content*, and target-side *history*. These components form the *condition sequence*, with the relevant tracks from the original music as the *target sequence*. The model is trained to reconstruct the music from these components.

This underexploration is likely not due to the absence of pre-trained models or their representational power, but to the lack of training objectives that support flexible adaptation to the diverse requirements of arrangement tasks. While sequence-to-sequence fine-tuning is conceptually a natural solution, it relies on parallel datasets—collections of the same piece arranged for different instrumentations (e.g., orchestra and piano)—which are extremely scarce. Even when adopting some level of internal parallel data [29, 30], these datasets constrain output directions to fixed mappings (e.g., orchestra to piano). In contrast, real-world arrangement demands far greater flexibility: the ability to transform arbitrary input instrumentations into arbitrary targets—i.e., support any-to-any transformations.

To address these limitations, we propose a unified framework for symbolic music arrangement that enables a pre-trained symbolic generative model to be fine-tuned across diverse tasks through a single self-supervised training pipeline (Figure 1). This facilitates the transfer of musical knowledge from generative pre-training to improve arrangement quality, while reducing the need for task-specific model design. To achieve this, we designed a context-aware, segment-level reconstruction objective: the model reconstructs multitrack music from its disentangled components, including content (notes executed) and style (instrumentation). We further observe that strictly time-ordered tokenizations (e.g., REMI+ [31]) introduce redundancy and fragment track content (detailed in §2.2), hindering instrument-level control and modeling. To address this, we propose a structured tokenization scheme that relaxes global time ordering while promoting track-wise continuity, enabling more consistent encoding across musical contexts. Our main contributions are as follows:

- We propose **a unified framework for symbolic music arrangement** that supports flexible instrumentation transformation across multiple typical arrangement scenarios, all without requiring parallel data. Central to our approach is a shared reconstruction objective defined over token-level disentangled note properties, which enables a single generative model to adapt through lightweight fine-tuning.

- To support effective learning under the proposed objective, we introduce an **efficient and modeling-friendly tokenization scheme** for multitrack music that produces shorter sequences with lower complexity, facilitates instrument-level control and modeling that are important for arrangement performance. It further reduces note-level perplexity in unconditional generation, indicating potential utility beyond arrangement tasks.

- Instantiated with a small model and modest-scale pre-training, our system **outperforms task-specific SOTA baselines** on three representative arrangement tasks, each corresponding to a distinct scenario—band arrangement (reinterpretation), piano reduction (simplification), and drum arrangement (additive generation)—in both objective and subjective evaluations.

## 2 Related Work

### 2.1 Automatic Arrangement in Multitrack Symbolic Music

Symbolic music arrangement encompasses a variety of tasks such as chord progression generation [38], orchestration from lead sheets [34], and instrumentation transfer [42]. In this work, we focus on adapting multitrack music to new instrumentations—a representative setting that requires fine-grained control over musical content and instrumentation.

Early supervised learning approaches rely on parallel datasets (e.g., piano-to-orchestra [5], band-to-piano [29, 30]), which are expensive to construct and inherently constrain the direction of arrangement. Classification-based methods [9] approach the arrangement problem by learning to assign instrument labels to each note from a dataset with fixed instrumentation. However, such models are limited in expressiveness, as they cannot modify musical content—e.g., adding, removing, or altering notes—to suit a target instrument combination. Moreover, their reliance on fixed instrument sets hinders generalization to unseen combinations.

Recent self-supervised methods [41, 42] avoid these constraints by guiding generation with pre-defined or autoregressively modeled high-level descriptors (e.g., pitch histogram, note density). While effective for maintaining stylistic coherence, they separate style modeling from content realization. As a result, instrument playing styles are either fixed or modeled independently of input variations. Since musical material often changes between sections, this decoupling can degrade fidelity—causing arrangements to fail to reflect core aspects such as melody or texture, resulting in noticeable perceptual divergence. This suggests a more fidelity-oriented approach: integrating style modeling into the generation process to allow dynamic adaptation to input content.

Additionally, infilling-based models such as Composer's Assistant [20, 19], although capable of handling additive generation scenarios such as drum arrangement, do not preserve the music essence of the original composition and are thus unsuited for reinterpretation and simplification.

### 2.2 Symbolic Music Tokenization

Transformer-based symbolic music modeling requires converting musical data into token sequences, a nontrivial task due to music's inherent multi-stream structure, i.e., multiple instruments playing concurrently. ABC-based notations [23] convert staff-based music into text-like symbolic representations, which are well suited for classical sheet music. While for comtemporary music, many existing tokenization schemes operate on MIDI files and adopt a linearized, note-event-based encoding where each musical note is represented as a tuple of attribute tokens (e.g., onset, pitch, duration, velocity, instrument). Some use absolute timing [13, 12, 39, 10], while others use metric durations [26, 14, 31]. Among them, the REMI representation [14], originally designed for single-track music, has been extended to REMI+ [31] for flexible tokenization of multitrack music, by associating each note with an instrument token.

However, existing time-ordered tokenization schemes such as REMI+ face structural limitations that hinder track-wise control and modeling. By flattening multitrack music into a strictly time-ordered sequence, REMI+ interleaves events from different instruments, resulting in **content fragmentation**. As illustrated in Figure 3a, notes from the same instrument (e.g., i-29, distorted electric guitar, in orange) are interrupted by those from concurrent instruments (e.g., i-80, synth lead, in red). This leads to two major issues. First, the lack of structured syntax makes it difficult to delineate track boundaries, hindering enforcement of user-specified instrument constraints and often leads to spurious instruments in arrangement outputs. Second, identical track-wise content can be tokenized differently depending on concurrent instrument activity. This context sensitivity reduces the repetition of typical per-instrument patterns in the training data—a key factor for learning accurate "instrument syntax" determined by physical constraints and idiomatic playing patterns of each instrument, important for arrangement quality. While some recent work explores vocabulary-level compression (e.g., Byte Pair Encoding [11]) to reduce sequence length—an approach orthogonal and complementary to tokenization scheme design—it does not address the structural issues discussed above.

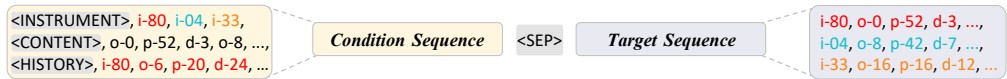

Figure 2: An example of the tokenized sequence for the band arrangement task. Special tokens <SEP>, <INSTRUMENT>, <CONTENT>, and <HISTORY> are used to separate different components. Tokens starting with o-, i-, p-, d- represents the onset, instrument ID, pitch, and duration of notes.

## 3 Method

Our goal is to fine-tune a single pre-trained symbolic music generative model across diverse arrangement tasks via a unified pipeline. The core is a segment-level reconstruction objective (§3.1) over token-level disentangled representations of style (instrumentation) and content (musical notes). To support track-wise control and modeling, we introduce a structured tokenization scheme (§3.2) that reduces content fragmentation and enhances learning under this objective.

### 3.1 Reconstruction from Token-Level Disentangled Multi-Streams with Context Awareness

Symbolic music offers a unique opportunity for token-level disentanglement: each note is represented as a list of semantically independent tokens, each describing a unique property of the note (e.g., onset, duration, pitch, instrument), allowing content (what is played) and instrumentation (by whom it is played) to be explicitly separated. This level of structural redundancy is rare in natural language, where sub-word tokens are atomic.

Building on this observation, we formulate arrangement as a self-supervised, segment-level reconstruction task. The input music is decomposed into three token streams—instrument, content, and preceding context—and a pre-trained symbolic music generative model is fine-tuned to reconstruct the desired tracks of the multitrack music from these components.

Let $y^{(t)}$ denote the $t$-th *segment* of a music piece. The fine-tuning objective is defined as:

$$\mathcal{L}(\theta) = -\log p_\theta\big(\mathcal{T}_{\text{task}}(y^{(t)}) \,\big|\, \text{I}(\mathcal{T}_{\text{task}}(y^{(t)})),\, \text{C}(\mathcal{S}_{\text{task}}(y^{(t)})),\, \mathcal{T}_{\text{task}}(y^{(t-1)})\big), \tag{1}$$

where $\theta$ represents the model parameters, $\text{I}(\cdot)$ and $\text{C}(\cdot)$ extract instrument and content conditions, $\mathcal{S}_{\text{task}}$ and $\mathcal{T}_{\text{task}}$ are filters that select task-specific source and target tracks respectively (detailed in $ 3.3), and $y^{(t-1)}$ provides target-side history. Equation 1 is implemented using a standard next-token prediction objective on the sequence [condition]<sep>[target] (Figure 2), with cross-entropy loss computed only on the target subsequence.

**Instrument condition.** The instrument condition specifies the desired instruments to be used in the segment. During training, it includes all instrument tokens from the target sequence. Their order encodes pitch register relationships across tracks: instruments with higher average pitch are placed earlier. At inference, users can specify arbitrary desired instruments (instrument control) and define their relative pitch register ordering (voice control).

**Content condition.** The content condition is derived from the original composition, encoded as a *content sequence* without instrument information. To obtain it, we remove all instrument tokens from the original multitrack sequence, sort notes by onset time, then by descending pitch, and merge duplicates. This yields a clean time-ordered note sequence conveying what is played, independent of how or by whom.

**History condition.** The history condition provides musical context from the preceding segment, encouraging inter-segment coherence at inference time. During training, it is provided via teacher-forcing with the complete tokenized previous segment; at inference, it is autoregressively generated. This mechanism helps maintain coherent arrangement style across segments, especially important for long-form (e.g., song-level) arrangement scenarios.

During training, the instrument, content, and history conditions are derived from the same music being reconstructed. The model thus learns to reinterpret musical content with various instrument combinations under specific contexts, enabling diverse arrangement behaviors.

**o-0** i-26 p-60 d-26 **o-0** i-33 p-36 d-23 **o-0** i-29 p-36 d-10 **o-12** i-29 p-36 d-12
**o-18** i-80 p-74 d-14 **o-18** i-29 p-48 d-12 **o-24** i-29 p-36 d-8 **o-30** i-29 p-52
d-11 **o-36** i-80 p-76 d-11 **o-36** i-29 p-36 d-10 **o-42** i-29 p-52 d-7 b-1

**i-80** o-18 p-74 d-14 o-36 p-76 d-11 **i-26** o-0 p-60 d-26 **i-29** o-0 p-36 d-10
o-12 p-36 d-12 o-18 p-48 d-12 o-24 p-36 d-8 o-30 p-52 d-11 o-36 p-36 d-10
o-42 p-52 d-7 **i-33** o-0 p-36 d-23 b-1

(a) REMI+ tokenization, demonstrated with REMI-z vocabulary.

(b) A REMI-z bar sequence containing four track sequences.

Figure 3: REMI+ and REMI-z tokenization for the same bar. Contents of the same instruments are highlighted with the same color. See Appendix A for complete vocabulary.

## 3.2 REMI-z: Tokenizing Multitrack Music with Track-Wise Continuity

As discussed in Section 2.2, strictly time-ordered tokenization schemes suffer from content fragmentation, a structural limitation that hinders arrangement performance. To mitigate this issue, we propose *REMI-z*, a tokenization scheme that heuristically prioritizes track-wise continuity over global time ordering. Specifically, REMI-z processes MIDI files into a list of *bar sequences*, each containing multiple *track sequences*, where each track corresponds to a unique instrument. Within each track, note events are sorted by their onset position, then by descending pitch, and grouped under a single instrument token. Tracks within a bar are then ordered by their average pitch (high to low), forming a bar-level token sequence that ends with a special end-of-bar token. The complete vocabulary and tokenization examples are shown in Appendix A.

This zig-zag organization—reflected in the REMI-z name—offers several modeling advantages: (1) instrument-wise content is locally contiguous (Figure 3b), reducing content fragmentation and enabling clear track boundaries; (2) sequence length is reduced by eliminating redundant instrument tokens; (3) temporal structure is preserved both within individual track sequences and between bars.

We adopt REMI-z to tokenize all data for both pre-training and fine-tuning. Beyond improving arrangement performance, we also observe that REMI-z produces sequences with lower information entropy and, when used for unconditional generative training, leads to better note-level modeling (see § 5.4). This suggests that its track-continuity design not only benefits arrangement tasks, but may also support general symbolic music modeling.

## 3.3 Task Instantiations

We evaluate our method on three representative music arrangement tasks that reflect typical arrangement scenarios, each assessing different capabilities of the model: band arrangement (reinterpretation), piano reduction (simplification), and drum arrangement (additive generation).

**Band Arrangement.** This task evaluates the model's ability to reinterpret an existing piece using arbitrary combinations of pitched instruments. The model must learn the properties and idiomatic playing styles of various instruments to reallocate or generate notes appropriately. Both $\mathcal{S}_{\text{task}}$ and $\mathcal{T}_{\text{task}}$ are identity mappings (i.e., $\mathcal{S}_{\text{task}}(y) = \mathcal{T}_{\text{task}}(y) = y$), and drum tracks are removed from the input. To encourage creative rewriting, we randomly remove a subset of tracks from the content condition $C(\mathcal{S}_{\text{task}}(y^{(t)}))$ during training, while ensuring the melody's content is preserved (detailed in Appendix B.2). Duration tokens are also removed from the content stream, allowing the model to infer track-specific note durations, i.e., articulations, suitable for interpreting the content under desired instrumentations. Segment length is set to 1 bar. This setup resembles [42], but differs in its more challenging setting: no track-wise style priors are available to guide the generation process.

**Piano Reduction.** This task simplifies ensemble music into solo piano accompaniment, aiming to preserve key harmonic and textural elements while ensuring pianistic playability. The $\mathcal{T}_{\text{task}}$ selects piano tracks, while $\mathcal{S}_{\text{task}}$ is the identity mapping. To ensure the reduction is meaningful, we filter training data to keep only segments where the piano part is sufficiently prominent (covering >40% of the pitch range). Drum tracks are removed from input, and the segment length is set to 1 bar. The setup is similar to [30], but uses original MIDI piano tracks as targets instead of human-composed reductions, and does not include difficulty-level conditioning.

**Drum Arrangement.** The goal of this task is to generate a drum track for music that lacks one. Here, $\mathcal{S}_{\text{task}}$ extracts all pitched-instrument tracks, while $\mathcal{T}_{\text{task}}$ extracts the drum track. We use 4-bar segments since drum patterns often span across multiple bars. The model must recognize the underlying groove and phrase boundaries of the source music, enhance them with coherent and stylistically appropriate rhythmic patterns, and ensure proper transitions across segments—requiring a stronger understanding

of rhythmic ideas and structural organization. This track infilling setup is similar to [19], but the model is not allowed to condition on future segments.

## 4 Experiments

### 4.1 Implementation Details

Our model, an 80M-parameter decoder-only Transformer, has a hidden dimension of 768, 12 layers, 16-head attention, and a context length of 2048 tokens (around $8\times$ the longest bar in our dataset). The model first undergoes a standard next-token-prediction pre-training, and then was fine-tuned with the proposed objective. Pre-training used four RTX A5000 GPUs (batch size 12, 1 epoch), while fine-tuning used a single A40 GPU (variable batch size, 3 epochs). Pre-training adopted the Los Angeles MIDI dataset [15] (405K MIDI files, 4.3B tokens after REMI-z tokenization, 2% validation split) and fine-tuning was done with Slakh2100 [21] (1,289 training, 270 validation, 151 test MIDI files), featuring 34 pitched instruments and drums, with $\geq 4$ tracks per piece. Detailed hyperparameter settings are in Appendix B.5.

### 4.2 Baseline Models

For each task, we compare our model against a state-of-the-art (SOTA) task-specific baseline. For band arrangement, we adopt **Transformer-VAE** from [42], the strongest previously reported model for multitrack arrangement without assumptions on track type or number. It combines Transformer-based long-term and inter-track modeling with a VQ-VAE generation module. For piano reduction, we compare with [30] (**UNet**), the most recent work in this area. For drum arrangement, we adopt Composer's Assistant 2 (**CA v2**) [19], a SOTA track infilling model capable of handling multitrack inputs and generating drum outputs. Existing drum-specific models (e.g., [2], [6]) are unsuitable for our setting, as they assume a single melody or instrumental track input rather than general multitrack conditioning. Baseline's implementation details are in Appendix B.4.

To demonstrate the impact of generative pre-training, an ablation variant of our model without pre-training (**w/o PT**) is used as a baseline. Additionally, we include simple rule-based baselines as non-learning references, designed to provide naive solutions with minimal musical knowledge, helping to contextualize the difficulty of arrangement tasks. For band arrangement, **Rule-Based** distributes notes evenly by pitch across instruments, serving as a naive reinterpretation strategy. For piano reduction, we use **Rule-F** (a flattened multitrack where the piano plays all notes), which reflects an overcomplete reduction prioritizing coverage, and **Rule-O** (the original piano track), which provides a playability-guaranteed but musically incomplete reduction. For drum arrangement, the original drum track (**Ground Truth**) is included anonymously in the human evaluation as an upper bound on perceptual scores.

### 4.3 Objective Evaluation

Objective metrics measure similarity between model outputs and target sequences, assuming closer resemblance to human-created music indicates higher naturalness and musicality. Following [19, 30], we use *note-level F1* to measure similarity between model outputs and target sequences. Specifically, we compute **Note F1** (correct onset and pitch) and **Note$_i$ F1** (additional correct instrument prediction), both under 16th-note quantization for fair comparison with baselines.

For piano reduction and drum arrangement, the same models are used in objective and subjective evaluations. For band arrangement, models are separately trained without random track deletion to ensure deterministic outputs. Baseline models are also modified by excluding its prior model to remove long-term context hints, ensuring evaluation fairness.

In addition to the modifications described above, we introduce three task-specific metrics for band arrangement. First, **Instrument Intersection over Union (I-IoU)** evaluates the accuracy of instrument control. Second, **Voice Error Rate (VER)** measures the similarity in voice features between the generated output and the reference, reflecting how well the model follows the voice conditions specified by instrument token ordering. Third, **Melody F1 (Mel F1)** computes the Note F1 score on melody tracks, estimated as the tracks with the highest average pitch in the output and reference, to

Table 1: Objective evaluation results for the band arrangement task. Statistical significance is indicated as follows: $*$ for $p < 0.05$, $\dagger$ for $p < 0.01$, and $\ddagger$ for $p < 0.001$.

| Model | I-IOU ↑ | VER ↓ | Note F1 ↑ | Note$_i$ F1 ↑ | Mel F1 ↑ |
|---|---|---|---|---|---|
| Transformer-VAE | 97.5 | 35.0 | 49.5 | 40.0 | 24.7 |
| Transformer w/ REMI+ | 95.0 | 18.2 | 94.4 | 76.0 | 68.8 |
| Transformer w/ REMI-z | $\ddagger$99.5 | $\ddagger$9.9 | $\ddagger$**97.8** | $\ddagger$77.5 | $\ddagger$77.8 |
| + Pre-training (Ours) | *99.8* | **7.6** | *97.5* | **87.0** | **84.5** |
| − voice | 99.6 | 17.6 | 97.2 | *84.3* | *81.5* |
| − history | **100.0** | *9.0* | 97.6 | 77.4 | 79.4 |

assess how well the original melody is preserved—an important factor for perceived fidelity. Detailed definitions and computation procedures are provided in Appendix C.1.

Among all tasks, band arrangement is a strong testbed for evaluating controllability and generalizability because it requires the highest flexibility without assumptions on target instrument types or counts. Hence it serves two additional purposes: (1) to validate key design choices in our fine-tuning objective, particularly the use of voice-aware instrumentation and segment-level history conditioning; and (2) to prove the effectiveness of the proposed tokenization schemes in arrangement task. When comparing tokenization schemes, statistical significance is computed by Wilcoxon signed rank test [36]. For a broader analysis of tokenization's impact on unconditional generation, see § 5.4.

## 4.4 Human Evaluation

To complement similarity-based objective metrics and assess perceptual quality and creativity, we conducted human evaluations. Full-piece arrangements were generated by all models and evaluated on a 5-point scale (1: very low, 5: very high). For band arrangement, models were tested across three instrument combinations with different complexity: string trio (3 tracks), rock band (4 tracks), and jazz band (7 tracks). Three metrics were used across band, piano, and drum arrangement tasks: **Coherence**, which evaluates the natural flow of the arrangement and the consistency of each instrument's playing style throughout the piece; **Creativity**, which assesses the degree of innovation in the arrangement under the constraints of the music's content and style; and **Musicality**, which measures the overall musical appeal and aesthetic quality of the arrangement. Further details on the metrics, ensemble settings, questionnaire, and evaluation process are in Appendix D.

Task-specific metrics were introduced for the distinct evaluation needs of each arrangement scenario. For band arrangement, **Faithfulness** measures resemblance to the original in melody and overall feel, while **Instrumentation** assesses the appropriateness of each instrument's role within the ensemble and their harmony. For piano reduction, **Faithfulness** is also adopted but without melody preservation requirements, and **Playability** assesses the feasibility of the generated contents played by human pianists. For drum arrangement, **Compatibility** measures how well the drum track blends with other instruments, and **Phrase Transition** assesses the smoothness of transitions between musical phrases. We report mean and standard error of mean in result tables. Significance tests were conducted between our model and the SOTA baselines using within-subject (repeated-measures) ANOVA [28].

## 5 Results

### 5.1 Band Arrangement

#### 5.1.1 Objective Evaluation

**Tokenization impact.** Table 1 shows that model adopted REMI-z significantly outperforms that adopted REMI+ across all objective metrics. It enables stronger instrument control (I-IOU: 99.5% vs. 95.0%) and voice control (VER: 9.9% vs. 18.2%), aligning outputs better with user-specified conditions. The improvement in Note F1 (+6.7%) confirms higher reconstruction quality, while the gain in Note$_i$ F1 (+1.9%) highlights enhanced instrument-wise modeling. Furthermore, the substantial increase in Mel F1 (+9.0%) suggests REMI-z enable the model to better identify the melody components from content sequence, which is improtant for arrangement fidelity.

Table 2: Band arrangement subjective evaluation results. Fa., Co., In., Cr., and Mu. represent Faithfulness, Coherence, Instrumentation, Creativity, and Musicality, respectively.

| Model | Fa. ↑ | Co. ↑ | In. ↑ | Cr. ↑ | Mu. ↑ |
|---|---|---|---|---|---|
| Rule-Based | *3.46*±0.14 | *3.05*±0.13 | *2.89*±0.14 | *3.00*±0.12 | *3.07*±0.13 |
| Transformer-VAE | 2.65±0.10 | 2.70±0.11 | 2.72±0.12 | *3.00*±0.13 | 2.72±0.11 |
| Ours | ‡**3.77**±0.13 | ‡**3.47**±0.15 | ‡**3.49**±0.16 | ***3.40**±0.13 | ‡**3.47**±0.14 |
| w/o PT | 3.19±0.13 | 2.82±0.13 | 2.86±0.14 | 2.93±0.12 | 2.75±0.13 |

Table 3: Piano reduction results. The Pl. represents Playability score.

| Model | F1 ↑ | Fa. ↑ | Co. ↑ | Pl. ↑ | Cr. ↑ | Mu. ↑ |
|---|---|---|---|---|---|---|
| Rule-F | - | **3.93**±0.13 | *3.59*±0.13 | 3.14±0.13 | *2.96*±0.13 | *3.34*±0.14 |
| Rule-O | - | 2.75±0.13 | 3.49±0.13 | **4.07**±0.12 | 2.62±0.14 | 2.96±0.14 |
| UNet | 58.3 | 2.97±0.13 | 2.90±0.15 | 3.47±0.13 | 2.82±0.13 | 2.78±0.13 |
| Ours | **85.5** | ‡*3.63*±0.13 | ‡**3.64**±0.13 | *3.86*±0.13 | *3.14*±0.12 | ‡**3.48**±0.14 |
| w/o PT | *78.4* | 2.25±0.13 | 2.58±0.16 | 3.29±0.15 | 2.67±0.15 | 2.26±0.14 |

**Pre-training benefit.** Pre-training brings clear gains by transferring musical knowledge useful for arrangement. It improves three key aspects: (1) lower VER ($-2.3\%$), indicating more effective voice control; (2) higher $\text{Note}_i$ F1 ($+9.5\%$), reflecting enhanced instrument-wise content modeling; and (3) the highest Mel F1 (84.5%), indicating stronger melody preservation. Probing analysis (Appendix E) further shows that pre-training strengthens the alignment between token embeddings and musical concepts useful for arrangement, such as instrument roles and chord progression.

**Comparison with Transformer-VAE.** Our model outperforms Transformer-VAE by a wide margin. In particular, it achieves much higher Note F1 (97.5% vs. 49.5%), $\text{Note}_i$ F1 (87.0% vs. 40.0%), and Mel F1 (84.5% vs. 24.7%), highlighting the advantage of our context-aware, content- and instrument-conditioned generation approach over the latent inference used in Transformer-VAE.

**Ablation: voice and history conditioning.** Removing voice information from instrument conditions degrades voice control (VER: $+10.0\%$), demonstrating the effectiveness of our voice control method. It also lowers per-instrument F1 ($\text{Note}_i$: $-2.7\%$) and Mel F1 ($-3.0\%$), indicating that voice-order information serves as a useful hint for inferring instrument roles during arrangement. Excluding history conditioning results in even larger drops in $\text{Note}_i$ F1 ($-9.6\%$) and Mel F1 ($-5.1\%$), validating that temporal context facilitates accurate reconstruction, laying the foundation for coherent song-level arrangement.

### 5.1.2 Human Evaluation

**Strong subjective performance.** As shown in Table 2, Our model achieves the highest scores across all subjective metrics than all baselines, demonstrating its ability to generate coherent, stylistically appropriate, and musically appealing arrangements while preserving core musical essence.

**Transformer-VAE underperforms.** It lags significantly behind our model in every subjective criterion. Notably, its lower Faithfulness score (-1.12) reflects difficulty in preserving core musical content, while low ratings in Instrumentation (-0.77) and Coherence (-0.77) suggest weaker track-wise modeling and non-idiomatic instrument usage, as well as insufficient coherence between segments. Overall Musicality (-0.75) also falls below our model. It even scores lower than the rule-based baseline, though the latter suffers from inherent limitations in musicality and coherence.

**Pre-training improves quality.** Removing pre-training reduces scores across all metrics, with notable drops in Faithfulness (-0.58) and Musicality (-0.72), reinforcing the importance of musical knowledge transfer from generative pre-training for content retention and overall perceptual quality.

### 5.2 Piano Reduction

**Best overall quality.** As shown in Table 3, our method obtains the highest ratings in F1 (85.5%), Coherence (3.64), Playability (3.86), and Musicality (3.48), while also maintaining strong scores in Faithfulness (3.63) and Creativity (3.14). It significantly outperforms UNet across all subjective

Table 4: Drum arrangement results. Comp. and Tr. represent Compatibility and Phrase Transition score respectively.

| Model | F1 ↑ | Comp. ↑ | Co. ↑ | Tr. ↑ | Cr. ↑ | Mu. ↑ |
|---|---|---|---|---|---|---|
| Ground Truth | **100.0** | **4.31**±0.12 | **4.18**±0.10 | *3.36*±0.13 | *3.16*±0.12 | *3.78*±0.12 |
| CA v2 | 20.3 | 3.82±0.13 | *4.05*±0.12 | 2.86±0.12 | 2.58±0.11 | 3.19±0.12 |
| Ours | *79.3* | *3.91*±0.12 | 4.03±0.10 | ‡**3.77**±0.12 | ‡**3.27**±0.14 | †*3.57*±0.13 |
| w/o PT | 1.2 | 2.49±0.16 | 2.19±0.12 | 2.21±0.14 | 2.82±0.15 | 2.05±0.13 |

Table 5: Tokenization scheme comparison on unconditional generation.

| Tokenizer | $\bar{T}_{\text{bar}}$ ↓ | $\bar{T}_{\text{note}}$ ↓ | $\bar{H}_{\text{bar}}$ ↓ | $\text{PPL}_{\text{note}}$ ↓ | $\text{PPL}_{\text{token}}$ ↓ |
|---|---|---|---|---|---|
| REMI+ | 225.91 | 4.03 | 41.68 | 116.20 | **3.00** |
| REMI-z (Ours) | **151.68** | **2.77** | **29.43** | **84.11** | 4.50 |

metrics, with especially large margins in Faithfulness (+0.66), Coherence (+0.74), and Musicality (+0.70)—indicating better content preservation, cross-segment continuity, and overall musical quality. Pre-training again proves essential: without it (w/o PT), performance drops notably across all metrics.

**Balanced fidelity and playability.** Compared to rule-based methods, our model achieves a better trade-off between fidelity and playability. Rule-F preserves original content well (Faithfulness: 3.93) but results in poor playability (3.14), while Rule-O achieves the highest playability (4.07) but at the expense of faithfulness (2.75). In contrast, our method produces reductions with balanced Faithfulness (3.63) and Playability (3.86).

### 5.3 Drum Arrangement

**Best subjective quality among learned models.** As shown in Table 4, our model outperforms CA v2 in all subjective metrics except Coherence (4.03 vs. 4.05), and significantly improves Creativity (3.47 vs. 2.58) and Phrase Transition (3.27 vs. 2.86), which are essential for engaging, structurally-aware drum arrangements. It also achieves the highest Musicality score (3.57), closely approaching ground truth (3.78). Again, the w/o PT variant performs poorly across all metrics.

**Improved phrasing and variation.** Compared to CA v2, which often repeats similar drum patterns throughout a piece, our model produces more varied and context-aware rhythms that better reflect musical phrasing. The higher Phrase Transition score (+0.91) supports this observation, demonstrating the model's ability to capture structural changes in music during generation.

### 5.4 Tokenization Efficiency and General Modeling Advantages

Additionally, we evaluate REMI-z against REMI+ to assess its efficiency and effectiveness for generative modeling of symbolic multitrack music. We report several metrics to evaluate compactness of tokenization schemes and their unconditional modeling performance: 1) **average tokens per bar** ($\bar{T}_{\text{bar}}$), 2) **average tokens per note** ($\bar{T}_{\text{note}}$), 3) **Shannon entropy of bar-level token sequences** ($\bar{H}_{\text{bar}}$), and 4) **note-level perplexity** ($\text{PPL}_{\text{note}}$). Note that $\text{PPL}_{\text{note}}$ is the aggregated conditional probability of all note attribute tokens, normalized by the number of notes, enabling fairer comparison across tokenization schemes. Calculations are detailed in Appendix C.2.

**Compactness.** Tokenizing Slakh2100 with REMI-z yields a 32.9% reduction in sequence length per bar (151.68 vs 225.91) and fewer tokens per note (2.77 vs 4.03), effectively reducing training and inference computational costs.

**Representational Simplicity.** REMI-z also produces lower bar-level Shannon entropy (29.43 vs 41.68 bits/token) on Slakh2100 compared to REMI+, despite representing the same underlying musical content. This indicates reduced information redundancy, and suggests that REMI-z sequences consist of more predictable tokens, potentially facilitating the learning of generative models.

**Note-Level Modeling Advantage.** We train unconditional generation models on Slakh2100 with both tokenizations and compare note-level perplexity with the same architecture as our arrangement models. The model adopting REMI-z achieves substantially lower note-level perplexity (84.11 vs.

116.20), indicating better modeling of complete musical notes. Since notes form the atomic units of music, improvements in note-level modeling directly contribute to lower uncertainty at modeling bars and full compositions, suggesting utility in general symbolic music modeling beyond arrangement tasks. While REMI+ slightly outperforms in token-level perplexity, this does not translate to better modeling of higher-level structures.

# 6 Conclusion

We presented a unified framework for automatic music arrangement, centered around a reconstruction objective that enables diverse arrangement tasks without requiring parallel data, enabling knowledge transfer from generative pre-trained symbolic music models for arrangement tasks. Our approach also integrates a structured tokenization scheme, REMI-z, which convert multitrack music to compact and easy-to-model token sequences. Experimental results on band arrangement, piano reduction, and drum arrangement show that our method consistently outperforms task-specific baselines in both objective and subjective evaluations. These results suggest the potential of our framework as a general solution for symbolic music-to-music transformation.

# Acknowledgements

The authors would like to thank Chenfei Kang for providing valuable resources that supported the validation of the proposed idea, and Yisong Miao for insightful comments on training and coherence of language models, which substantially benefited this work.

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

# A  REMI-z Tokenization

## A.1  Vocabulary

Table 6 presents the complete vocabulary of our proposed REMI-z tokenizer, detailing the value ranges for each token type. For instrument tokens, the values correspond to MIDI program numbers (0–127), with 128 specifically designated for the drum set. For pitched instruments, pitch token values directly map to MIDI pitch numbers, while drum set pitches are encoded as MIDI pitch + 128. Both position and duration tokens are quantized in units of a 48th note (one-third of a sixteenth note). Time signature and tempo tokens undergo specific quantization before token mapping; for detailed quantization rules, please refer to our code implementation. All tokens listed in the table were utilized during pre-training, including time signature and tempo tokens serving as bar-level properties, with example REMI-z sequences illustrated in Figure 4b.

During fine-tuning, we simplified the token set by excluding time signature and tempo tokens. This decision was supported by our analysis of the fine-tuning dataset (Slakh2100 [21]), where 94.8% of songs use 4/4 time signatures. Consequently, we restricted our fine-tuning to 4/4 songs and omitted time signature tokens, following practices in related works [35, 42]. We also excluded tempo tokens since tempo adjustments in digital audio workstations are typically handled as a global parameter, affecting only tempo tokens without altering the any other tokens. This simplification assumes that compositional and performance styles remain consistent across different tempos. However, this assumption may not hold for datasets with significant tempo variations. Therefore, for future research requiring time-signature- or tempo-specific characteristics, we recommend including these tokens during fine-tuning.

Velocity tokens were excluded from both pre-training and fine-tuning phases. This decision reflects our focus on compositional quality rather than performance naturalism, aligning with previous approaches in band and piano arrangement studies [42, 30]. However, if velocity information is deemed crucial for generation, our model architecture readily accommodates the addition of velocity tokens to each note without introducing content fragmentation issues, if the REMI-z note organization order is maintained.

## A.2  Example

We name our tokenization scheme *REMI-z* for its distinctive "zig-zag" encoding pattern for musical notes within each bar, as illustrated in Figure 4a. In this scheme, notes are encoded hierarchically: first grouped by tracks (instruments), then organized bar by bar. This track-first approach ensures notes from the same instrument remain clustered together, thereby enhancing the model's ability to learn instrument-specific patterns. The resulting REMI-z sequence is demonstrated in Figure 4b. In contrast, REMI+ [31] employs a column-wise encoding strategy, strictly ordering notes by their temporal positions. While this approach effectively captures global temporal relationships, it disperses notes from the same instrument throughout the sequence, potentially complicating instrument-specific pattern learning. We leverage such temporal ordering in our content sequence (excluding instrument tokens), as shown in Figure 5.

Table 6: REMI-z vocabulary with pitch token distinctions

| Meaning | Token | X's range |
|---|---|---|
| Instrument type | i-X | 0∼128 |
| Note's within-bar position | o-X | 0∼127 |
| Note's pitch (non-drum) | p-X | 0∼127 |
| Note's pitch (drum) | p-X | 128∼255 |
| Note's duration | d-X | 0∼127 |
| End of a bar | b-1 | - |
| Time signature | s-X | 0∼253 |
| Tempo | t-X | 0∼48 |

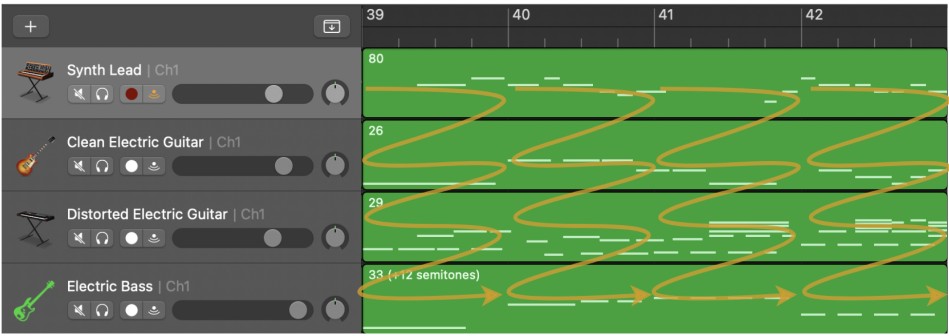

(a) A 4-bar musical segment with orange arrows illustrating the "zig-zag" encoding order of notes in REMI-z tokenization.

1st bar
(s-9 t-35) (optional)
**i-80** o-18 p-74 d-14 o-36 p-76 d-11 **i-26** o-0 p-60 d-26 **i-29** o-0 p-36 d-10 o-12 p-36 d-12 o-18 p-48 d-12 o-24 p-36 d-8 o-30 p-52 d-11 o-36 p-36 d-10 o-42 p-52 d-7 **i-33** o-0 p-36 d-23 b-1

2nd bar
(s-9 t-35) (optional)
**i-80** o-0 p-74 d-10 o-12 p-76 d-5 o-18 p-74 d-10 o-30 p-72 d-5 o-36 p-71 d-5 o-42 p-72 d-10 **i-26** o-0 p-67 d-13 o-18 p-67 d-12 o-30 p-67 d-11 o-42 p-64 d-11 **i-29** o-6 p-43 d-7 o-12 p-31 d-6 o-18 p-47 d-12 o-24 p-32 d-7 o-30 p-44 d-10 o-36 p-32 d-10 o-42 p-44 d-9 **i-33** o-0 p-43 d-17 o-24 p-44 d-9 o-36 p-44 d-8 b-1

3rd bar
(s-9 t-35) (optional)
**i-80** o-36 p-69 d-4 o-42 p-72 d-5 **i-26** o-6 p-64 d-11 o-18 p-60 d-17 **i-29** o-0 p-33 d-8 o-6 p-45 d-10 o-12 p-33 d-8 o-18 p-60 d-19 p-57 d-19 p-52 d-19 p-48 d-8 o-24 p-33 d-8 o-30 p-48 d-8 o-36 p-33 d-4 o-42 p-48 d-4 **i-33** o-0 p-45 d-8 o-12 p-45 d-8 o-24 p-45 d-8 o-36 p-45 d-8 b-1

4th bar
(s-9 t-35) (optional)
**i-80** o-0 p-76 d-5 o-6 p-74 d-10 o-18 p-74 d-5 o-24 p-72 d-5 o-30 p-72 d-4 o-36 p-74 d-5 o-42 p-72 d-13 **i-26** o-6 p-64 d-11 o-18 p-62 d-12 o-30 p-60 d-5 o-36 p-62 d-5 o-42 p-64 d-11 **i-29** o-0 p-40 d-8 o-6 p-52 d-9 o-12 p-40 d-8 o-18 p-62 d-12 p-59 d-12 p-55 d-12 p-52 d-11 o-24 p-40 d-8 o-30 p-60 d-4 p-52 d-11 o-36 p-62 d-5 p-40 d-8 o-42 p-60 d-13 p-57 d-13 p-53 d-13 p-48 d-8 **i-33** o-0 p-40 d-8 o-12 p-40 d-8 o-24 p-40 d-8 o-36 p-40 d-8 b-1

(b) The REMI-z sequence tokenized from the musical segment above. Track sequences are color-coded by instrument: synth lead (red), clean electric guitar (blue), distorted electric guitar (orange), and electric bass (purple).

Figure 4: An example of REMI-z tokenization.

1st bar
**o-0** p-60 d-26 p-36 d-23 **o-12** p-36 d-12 **o-18** p-74 d-14 p-48 d-12 **o-24** p-36 d-8 **o-30** p-52 d-11 **o-36** p-76 d-11 p-36 d-10 **o-42** p-52 d-7 b-1

2nd bar
**o-0** p-74 d-10 p-67 d-13 p-43 d-17 **o-6** p-43 d-7 **o-12** p-76 d-5 p-31 d-6 **o-18** p-74 d-10 p-67 d-12 p-47 d-12 **o-24** p-44 d-9 p-32 d-7 **o-30** p-72 d-5 p-67 d-11 p-44 d-10 **o-36** p-71 d-5 p-44 d-8 p-32 d-10 **o-42** p-72 d-10 p-64 d-11 p-44 d-9 b-1

3rd bar
**o-0** p-45 d-8 p-33 d-8 **o-6** p-64 d-11 p-45 d-10 **o-12** p-45 d-8 p-33 d-8 **o-18** p-60 d-19 p-57 d-19 p-52 d-19 p-48 d-8 **o-24** p-45 d-8 p-33 d-8 **o-30** p-48 d-8 **o-36** p-69 d-4 p-45 d-8 p-33 d-4 **o-42** p-72 d-5 p-48 d-4 b-1

4th bar
**o-0** p-76 d-5 p-40 d-8 **o-6** p-74 d-10 p-64 d-11 p-52 d-9 **o-12** p-40 d-8 **o-18** p-74 d-5 p-62 d-12 p-59 d-12 p-55 d-12 p-52 d-11 **o-24** p-72 d-5 p-40 d-8 **o-30** p-72 d-4 p-60 d-5 p-52 d-11 **o-36** p-74 d-5 p-62 d-5 p-40 d-8 **o-42** p-72 d-13 p-64 d-11 p-60 d-13 p-57 d-13 p-53 d-13 p-48 d-8 b-1

Figure 5: The content sequence obtained by applying the operator $C(\cdot)$ to the REMI-z sequence shown in Figure 4b.

## B  Implementation Details

### B.1  Key Normalization

Transposition-equivariance is a crucial property in symbolic music: a composition's musicality remains unchanged under global pitch shifts (uniform pitch adjustment across all notes). However, uneven key distribution in datasets can lead to data sparsity during training, potentially causing models to perform inconsistently across different keys. Two approaches address this issue: (1) data augmentation through systematic semitone transpositions $0, \pm1, \pm2, \cdots$ [19, 42], or (2) normalizing all songs to a common key (e.g., C major and A minor) [17]. We adopt the latter approach, implementing a modified version of [17]'s method. Our implementation uses a key dictionary mapping from each of the 24 keys (12 major and 12 minor) to a 12-dimensional binary vectors, where 1s indicate scale notes. Key detection is performed by computing dot products between these vectors and a song's pitch histogram, with the highest-scoring key determining the transposition needed to normalize to C major or A minor. Further details can be found in the code.

## B.2 Random Track Deletion

To encourage creativity in band arrangement and prevent the model from over-relying on direct copying notes from the input, we introduce a random track deletion mechanism during training. The intuition is that a well-trained model should be able to infer suitable instrumental content even if such content does not exist in the original music composition.

Concretely, given a content sequence with multiple instrument tracks, we randomly delete all notes that belongs to a subset of instruments before calculating the content sequences and feeding to the model. The number of instruments to delete is sampled from a Poisson distribution with $\lambda = \max(\lfloor |\mathcal{I}|/4 \rfloor, 1)$, where $\mathcal{I}$ is the set of non-melodic instruments in the content sequence. The melodic track is estimated by the track with highest average pitch, and does not involves in the track deletion to ensure the melody's content is preserved. Then, the deletion count is clipped to ensure at least one instrument remains. A corresponding number of instruments are uniformly sampled without replacement and all tokens associated with these instruments are removed from the input. Then the content sequence is calculated on this modified music sequence.

Importantly, the target sequence remains unchanged: it includes all instruments and notes removed from the input. This setup requires the model to reconstruct missing tracks based solely on the remaining musical context. In doing so, the model learns not only to replicate observed content but also to infer plausible notes for desired instruments in a given context. This can be treated as a music specific span infilling denoising objective for sequence-to-sequence learning [16], but we don't explicitly tell the model whether an input is masked and the location of masked spans.

## B.3 Instrument Quantization

**Band Arrangement**   While our tokenization scheme supports all MIDI program IDs, many instruments with different IDs share fundamental compositional properties, differing primarily in timbre (e.g., acoustic and electric pianos). To leverage these similarities and reduce instrument distribution sparsity in training data, during fine-tuning, we group similar instruments and assign them the lowest program ID within their group. The instrument grouping rules follows that of [21], leading to 34 different program IDs in total. For multiple tracks of the same instrument type, we merge their notes into a single track.

**Piano Arrangement**   For piano arrangement, we consolidate all piano-type instruments (MIDI program IDs 0-7, including both acoustic and electric pianos) into a single track to form the target sequence.

## B.4 Baseline Models

**Band Arrangement**   For band arrangement, we use the Transformer-VAE model from [42] as our baseline. For human evaluation, we utilized their official implementation without retraining. The model's generation module was trained on Slakh2100 [21] (the same dataset as our fine-tuning), and its prior model leveraged the larger Lakh MIDI Dataset [25], which encompasses Slakh2100. For objective evaluation, we retrained another version of the model on Slakh2100 that did not receive hints from the style prior model for a fair comparison.

**Piano Reduction**   Piano reductions can be categorized into two types: (1) piano accompaniments where the melody is delegated to a separate lead instrument, focusing solely on preserving harmony and texture (e.g., [41, 33]), which can be effectively handled using self-supervised methods; and (2) complete solo piano arrangements that additionally include the melody, requiring careful human arrangement and band-to-piano parallel datasets for supervised learning (e.g., [30, 29]). Due to the lack of open-source parallel datasets, we focus on the accompaniment arrangement task in this paper and use the term *piano reduction* interchangeably. However, our methodology is inherently flexible and can also handle the second type of reduction if parallel datasets become available. Additionally, exploring the potential for full solo piano reductions using unsupervised approaches remains a valuable direction for future research.

For comparison, we reimplemented the UNet baseline from [30] and trained it on Slakh2100 using our data preparation pipeline. Instead of adopting parallel data, we followed the same setting as our model, using the original composition as input and the piano track within the song as output. To

Table 7: Pre-train hyper-parameter setting.

| Hyperparameters | Values |
|---|---|
| seed | 42 |
| learning_rate | 0.0005 |
| weight_decay | 0.1 |
| train_batch_size | 12 |
| gradient_accumulation_steps | 8 |
| total_train_batch_size | 96 |
| optimizer | Adam with betas=(0.9,0.999) and epsilon=1e-08 |
| lr_scheduler_type | cosine |
| lr_scheduler_warmup_steps | 1000 |
| num_epochs | 1 |

ensure musical coverage, we selected only piano tracks that span more than 40% of the piece's pitch range.

Since hand-specific annotations (i.e., left and right hand separation) are unavailable, the entire piano track is generated as a single sequence rather than as separate streams for each hand. We also omitted octave shifts during input preprocessing for two reasons. The first is that, unlike their setup where human-composed piano references may intentionally transpose notes by octaves, our output consistently corresponds to a subset of the input notes without such shifts. The second is empirical: introducing octave shifts during training significantly degraded UNet's performance, reducing note-level F1 from 58.31 to 42.80 and diminishing perceptual quality.

**Drum Arrangement**   For drum arrangement, we employ Composer's Assistant v2 [19] as our baseline, specifically using v2.1.0 from their official repository[2]. We use the model without retraining since its original training data (Lakh MIDI Dataset [25]) encompasses our fine-tuning dataset (Slakh2100).

**Tokenization Comparison**   When comparing the effectiveness of REMI+ and the proposed REMI-z tokenization in §3.2, the model structure used is the same as our arrangement model, but with a simpler training objective—the standard left-to-right next-token-prediction. We did not use time signature tokens, tempo tokens, and velocity tokens in REMI+ for a fair comparison.

## B.5   Hyperparameter Settings

The model we adopted is a GPT-2 model comprising 12 Transformer decoder layers with a hidden size of 768. Our setup follows GPT's 12-layer, 768-dim embedding, 3072-dim inner states [24], and our experimental design is also in the same style: a single pretrained model and multiple fine-tuning tasks to show the efficacy of the pretrain–finetune paradigm. We used a slightly higher number of attention heads (16 instead of 12) based on the intuition that the interactions between different music tokens are more critical for musical quality than the value of individual note attributes. We detail the hyperparameter configurations used in our experiments below.

For pre-training, Table 7 summarizes the pre-training configuration, where hyperparameters were used as-is without optimization. The pre-training was implemented using `pytorch` and `transformers` frameworks on a Linux platform, while fine-tuning additionally utilized `lightning`.

For fine-tuning, we conducted a simple learning rate search over 1e-5, 5e-5, 1e-4, selecting the optimal value based on validation loss. This resulted in learning rates of 5e-5 for drum arrangement and 1e-4 for band arrangement and piano reduction. The batch sizes and context lengths were configured as follows: band arrangement used a batch size of 24 and context length of 768; piano reduction similarly adopted a batch size of 24 and context length of 768; drum arrangement employed a batch size of 8 and context length of 1536. Across all fine-tuning tasks, we used the AdamW optimizer with

---

[2]`https://github.com/m-malandro/composers-assistant-REAPER`

0.01 weight decay, incorporating a linear learning rate scheduler with 500-step warmup. Training spanned 3 epochs for band, piano, and drum tasks, with early stopping patience of 2 epochs. The best checkpoints were selected based on validation loss.

For the tokenization comparison, the model is trained from scratch without generative pre-training. The hyperparameter settings remain the same as those used for band arrangement fine-tuning, except for an increased number of epochs (5).

All experiments were conducted with a fixed random seed of 42.

## C Objective Metrics

In this section, we formally define the objective metrics used in the paper. All objective metrics are calculated at the segment level, and arithmetically averaged across the test set.

### C.1 Arrangement Evaluation

Most of our objective metrics are based on note-level F1 scores calculated on piano roll of 16-th note quantization. Given the extreme sparsity of note events in the track-wise piano roll (e.g., only 0.12% non-zero elements in the Slakh2100 dataset under 16th-note quantization), F1-based metrics are more suitable than accuracy-based metrics for similarity evaluation. Let $X = \{x_1, ..., x_n\}$ and $Y = \{y_1, ..., y_m\}$ be two sequences of note events, where each note event $e$ is defined as a tuple $e = (t, p)$ with onset time $t$ and pitch $p$. A note event $x_i \in X$ is considered to match $y_j \in Y$ if and only if:

$$|t_{x_i} - t_{y_j}| < \delta_t \text{ and } p_{x_i} = p_{y_j} \tag{2}$$

where $\delta_t$ is the temporal tolerance threshold (set to one 16th-note duration in our evaluation).

Let $M(X, Y)$ denote the set of matched note pairs between $X$ and $Y$. The **Note F1** score is defined as:

$$\text{Precision} = \frac{|M(X, Y)|}{|X|} \tag{3}$$

$$\text{Recall} = \frac{|M(X, Y)|}{|Y|} \tag{4}$$

$$\text{Note F1} = \frac{2 \cdot \text{Precision} \cdot \text{Recall}}{\text{Precision} + \text{Recall}} \tag{5}$$

The **Note$_i$ F1** extends this by considering instrument matching, where each note event becomes $e = (t, p, i)$ with $i$ representing the instrument. The matching criterion becomes:

$$|t_{x_i} - t_{y_j}| < \delta_t \text{ and } p_{x_i} = p_{y_j} \text{ and } i_{x_i} = i_{y_j} \tag{6}$$

Note$_i$ F1 is then computed using Equation 5 with this stricter matching criterion.

For **Melody F1 (Mel F1)**, given a multitrack piece with tracks $T = \{T_1, ..., T_k\}$, we first identify the melody track $T_m$ as:

$$T_m = \arg\max_{T_i \in T} \frac{1}{|T_i|} \sum_{e \in T_i} p_e \tag{7}$$

where $|T_i|$ is the number of notes in track $T_i$ and $p_e$ is the pitch of note event $e$. The Mel F1 score between output and target sequences is then:

$$\text{Mel F1} = \text{Note F1}(T_m^{out}, T_m^{tgt}) \tag{8}$$

For instrument control evaluation, **Instrument IoU (I-IoU)** measures the overlap between instrument sets. Let $\mathcal{I}^{out}$ and $\mathcal{I}^{tgt}$ denote the sets of instruments used in the output and target sequences respectively:

$$\text{I-IoU} = \frac{|\mathcal{I}^{out} \cap \mathcal{I}^{tgt}|}{|\mathcal{I}^{out} \cup \mathcal{I}^{tgt}|} \tag{9}$$

where an instrument is considered present if there exists at least one note event using it.

Finally, **Voice Error Rate (VER)** evaluates the similarity of voice arrangements between two multitrack compositions. For each piece, we derive an ordered voice sequence by:

1) Computing the average pitch $\bar{p}_i$ for each active instrument $i \in \mathcal{I}^{active}$:

$$\bar{p}_i = \frac{1}{|T_i|} \sum_{e \in T_i} p_e \tag{10}$$

2) Constructing a voice sequence $V$ by sorting instruments by descending average pitch:

$$V = [i_1, i_2, ..., i_n] \text{ where } \bar{p}_{i_k} \geq \bar{p}_{i_{k+1}} \text{ for } k = 1, ..., n-1 \tag{11}$$

The VER between output sequence $V^{out}$ and target sequence $V^{tgt}$ is:

$$\text{VER} = \frac{S + D + I}{N} \tag{12}$$

where $S$, $D$, and $I$ are the minimum number of substitutions, deletions, and insertions required to transform $V^{out}$ into $V^{tgt}$, and $N = |V^{tgt}|$ is the length of the $V^{tgt}$.

Among all the selected objective metrics, the Note$_i$ F1 is the most comprehensive and important one. To achieve a perfect Note$_i$ F1 score, the model need to perform perfectly in all evaluated aspects: instruments, voice, melody, note reconstruction and allocation.

## C.2 Tokenization Evaluation

**Average tokens per bar** ($\bar{T}_{\text{bar}}$) and **average tokens per note** ($\bar{T}_{\text{note}}$) are computed by first counting the total number of tokens and the number of bars or notes, respectively, across the entire dataset. Specifically, $\bar{T}_{\text{bar}}$ is obtained by dividing the total number of tokens by the number of bars, while $\bar{T}_{\text{note}}$ is obtained by dividing the total number of tokens by the number of notes.

**Bar-level Shannon Entropy** ($\bar{H}_{\text{bar}}$) is computed by measuring the Shannon entropy of each bar-level token sequence $H(X)$:

$$H(X) = -\sum_{i=1}^{N} P(x_i) \log_2 P(x_i), \tag{13}$$

where $X$ is the token distribution within a bar. Then we average it across all bars in the dataset to obtain $\bar{H}_{\text{bar}}$.

**Note-level Perplexity** (PPL$_{\text{note}}$) aggregates probabilities over all tokens representing a musical note (e.g., instrument, pitch, position, duration), and normalizes by the number of notes:

$$\text{PPL}_{\text{note}} = \exp\left(-\frac{1}{M} \sum_{j=1}^{M} \log P(n_j \mid n_{1:j-1})\right), \tag{14}$$

where $M$ is the number of notes in a bar.

Finally, **token-level Perplexity** (PPL$_{\text{token}}$) measures the standard autoregressive perplexity over all tokens. While commonly reported in language modeling, it serves as an auxiliary metric here, as it does not directly reflect the model's ability to model note events, which are basic units of music.

# D  Subjective Evaluation

## D.1  Subjective Metrics

These metrics are based on listeners' auditory experiences and their subjective feelings while listening to the music. Since they are subjective, they cannot be easily defined using mathematical equations. However, we detail the evaluation criteria through the prompt questions provided in the questionnaire, which participants answered after listening to the demos. Each metric is assessed on a 5-point scale, ranging from 1 (very low) to 5 (very high).

- **Coherence**: Does the arrangement flow naturally and smoothly? How consistent is each instrument's performance and style throughout the piece?

- **Creativity**: How creative is the arrangement while maintaining faithfulness and naturalness?
- **Musicality**: What is the overall musical quality?
- **Faithfulness** (band): How closely does the arrangement resemble the original piece in terms of melody and overall feel?
- **Faithfulness** (piano): How closely does the arrangement capture the overall feel of the original piece?
- **Instrumentation** (band-only): Does each instrument fulfill its appropriate role within the band, and do they harmonize effectively?
- **Playability** (piano-only): How well is the piece suited for piano? How likely is it that a human pianist could perform this accompaniment?
- **Compatibility** (drum-only): Is the drum beat compatible with the other instruments?
- **Phrase Transition** (drum-only): How effectively does the drum arrangement handle transitions between phrases?

## D.2 Human Evaluation Details

**Survey Design.** We conducted subjective evaluations using audio clips arranged by different models across three tasks: band arrangement, piano reduction, and drum arrangement. For band arrangement, we used out-of-domain test songs with novel compositions and instrument groups; for the other two tasks, test set songs were used to allow comparison with the original piano (Rule-O) or drum (ground truth) tracks. Each model was tasked with arranging the full song, and we selected a 15–30 second chorus phrase—the most representative segment—for evaluation. In total, we prepared 6 songs for band arrangement, 5 for piano reduction, and 5 for drum arrangement.

Each sample group includes the original music to be arranged and the anonymized outputs from all compared models (plus the ground truth for the drum task). Each sample is 8–16 bars long, rendered to audio in Cubase AI 13 using the default soundfont and the original MIDI's BPM. We collected 56, 73, and 77 evaluation groups for band, piano, and drum tasks respectively. The mean time spent per session was approximately 30 minutes. Figure 6 shows the sample survey interface and instructions.

**Participant Background** A total of 26 participants joined the study, 5 of whom work in the music industry. Among all evaluators, 73.1% have over 10 years of experience in music composition or performance. We conducted t-tests comparing these raters with those having ≤10 years of experience (using piano reduction ratings on our model), and found no significant differences in the means between the two groups ($p > 0.2$ across all metrics). This suggests that musical background did not systematically bias the results.

**Evaluation Protocol.** Participants first listened to the original music clip, followed by model outputs presented in random order. For each arranged clip, participants rated multiple aspects using a 5-point Likert scale, based on task-specific questions in the previous subsection. The evaluation groups were randomly distributed to participants to ensure unbiased feedback. Each rater evaluated a subset of groups, where one group = one song (input) with multiple model outputs. On average, raters evaluated 2.15 groups for band, 2.80 for piano, and 2.96 for drum — corresponding to 8.6, 14.0, and 11.8 samples respectively.

**Band Arrangement Settings.** To evaluate generalization, we designed three distinct instrumentation settings: (1) string trio (violin, viola, cello), (2) rock band (synth lead, clean electric guitar, distorted electric guitar, electric bass), and (3) jazz band (saxophone, violin, brass section, clean electric guitar, piano, string ensemble, electric bass). Each setting was applied to two different songs, covering six songs in total. The input content was drawn from both piano arrangements (3 songs) and band arrangements (3 songs), testing the model's ability to adapt to varying content and target instrument groups.

## D.3 Discussion on Evaluation Scheme

Our experiments primarily adopt a 5-point scale for subjective evaluation, which is the most commonly used protocol in prior music arrangement works [42, 35, 38]. However, we observe a growing trend toward using A/B testing to evaluate overall quality in the broader music generation literature [32,

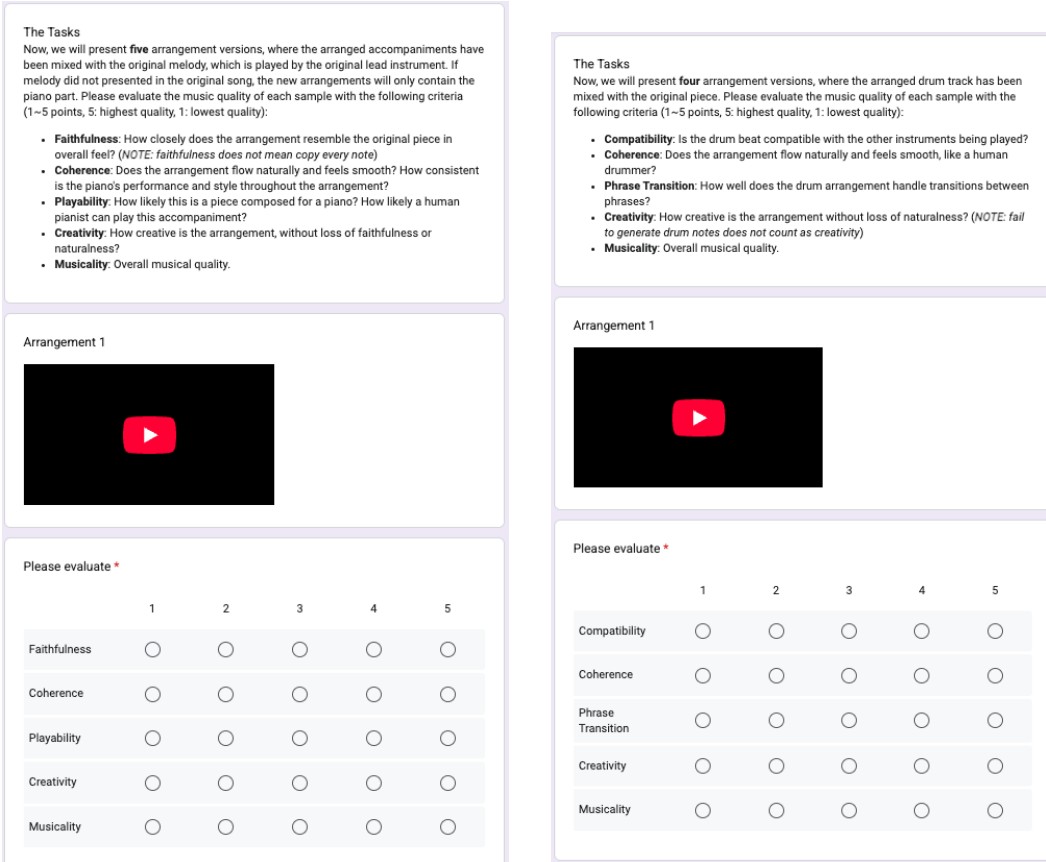

(a) An example of the piano reduction task.    (b) An example of the drum arrangement task.

Figure 6: Screenshots of survey pages and instructions of our online survey.

23, 37]. We encourage future research to consider combining both approaches to obtain more comprehensive and convincing evaluation results.

# E Probing Analysis of Pre-Training Impact

Knowledge probing techniques are used to explore what a model has learned within its hidden representations [27]. A widely adopted method is linear probing, where the pre-trained model is frozen and a simple linear classifier is trained on top of the hidden representations to predict specific properties, thereby revealing the extent to which particular knowledge is encoded in the model [1]. This technique is particularly useful in our context for evaluating what the model has internalized during pre-training.

We have shown in the paper that the proposed models outperform the baseline models that do not undergo the pre-traning stage on the generation quality. In this section, we further analyze the reason. Specifically, we conduct probing experiments to assess whether pre-training enhances the acquisition of musical knowledge to facilitate understanding content conditions in fine-tuning.

We focus on two probing tasks: (1) classifying instrument types from content sequences that contain only position, pitch, and duration tokens, and (2) recognizing chord progression sequences from content sequences. To determine whether instrument or chord information is linearly accessible within the model's sequence embeddings, we employ linear classifier probes. In these probing experiments, the average pooling of the Transformer's output embeddings across all tokens in a sequence is fed into the classifier. The model parameters are kept frozen, and only the linear classifiers are trained. We compare the knowledge captured by different models: a randomly initialized Transformer, a model that has undergone pre-training only (PT only), a model trained on the band arrangement task

Table 8: Instrument probing results.

| Model | Acc@1 | Acc@3 | Acc@5 |
|---|---|---|---|
| Random guess | 2.94 | 8.82 | 14.71 |
| Random initialized | 38.98 | 61.71 | 74.22 |
| FT only | 41.50 | 64.93 | 76.53 |
| PT only | **46.14** | **69.30** | **79.61** |
| PT + FT | *45.89* | *68.96* | *79.47* |

Table 9: Chord probing results.

| Model | Chord Root | | Chord quality | |
|---|---|---|---|---|
| | Acc@1 | Acc@3 | Acc@1 | Acc@3 |
| Random guess | 8.33 | 25.00 | 11.11 | 33.33 |
| Random initialized | 48.86 | 78.76 | 38.05 | 77.97 |
| FT only | 50.09 | 80.23 | 39.26 | 79.03 |
| PT only | **62.93** | **89.42** | **50.09** | **85.58** |
| PT + FT | *58.05* | *85.93* | *44.92* | *82.48* |

without pre-training (FT only), and a model that has undergone both pre-training and fine-tuning (PT + FT).

For the probing experiments, the batch size was set to 12 for chord probing and 64 for instrument probing. The learning rates were 5e-4 for chord probing and 1e-4 for instrument probing. Both experiments shared the same remaining hyperparameters: training for 10 epochs, using a linear learning rate scheduler, a 500-step warmup, and a weight decay of 0.01.

### E.1 Instrument Type Probing

In this task, we use a linear probe to estimate the instrument type from a single-track music sequence without providing instrument tokens. The goal is to predict which instrument is most likely to play the given note sequence. The performance is evaluated using top-1, top-3, and top-5 prediction accuracy metrics.

As shown in Table 8, a model initialized with random weights shows notable improvement after fine-tuning, suggesting that the ability to discern instrument styles is the requirement for performing well in music arrangement tasks. However, the gains in accuracy are modest, likely due to the limited number of training samples, which may constrain the model's ability to aquire such knowledge through fine-tuning alone. Interestingly, models that underwent pre-training exhibit the highest accuracy in predicting instrument types. This indicates that substantial knowledge of instrument styles can be acquired effectively during the pre-training phase. Moreover, the proposed models that are both pre-trained and fine-tuned (PT+FT) maintained high accuracy levels, demonstrating that the knowledge about instruments styles are useful for the arrangement task.

### E.2 Chord Progression Probing

In this task, we use linear probes to predict chord progressions from a 2-bar music content sequence without instrument tokens. Eight linear probes are trained simultaneously to predict the chord roots and qualities for a total of four chords (two chords per bar). This tasks is used to evaluate whether the model contains position-specific chord information. Similarly, we use top-1 and top-3 prediction acuracy metrics.

As shown in Table 9, the model with only fine-tuning (FT only) indicates a foundational grasp of chord knowledge for this complex music analysis tasks. However, similar to instrument prediction, the knowledge of recognizing chord progression does not significant gain, until the pre-training is also introduced into the model, confirming that pre-training establishes a robust basis for better understanding of content sequence, which may potentially help with the quality of arrangement.

# F Broader Impact

This work presents a unified framework for symbolic music arrangement using pretrained generative models, achieved by a reconstruction fine-tuning objective and a structured tokenization scheme. By enabling high-quality arrangement generation under flexible control, our approach has the potential to make music creation more accessible to non-experts, reduce the technical burden for composers, and support educational and assistive tools for learning music theory, orchestration, or instrumentation. It may also benefit creative professionals by streamlining workflows in game audio, film scoring, and digital content production.

Many other conditional generation tasks could similarly benefit from this training objective design. For instance, one can construct a conditional sequence derived from a partial or aspect-wise decomposition of the target, concatenate it with the original target tokens, and then perform segment-level generation using a pretrained model with historical context-awareness to fine-tune existing generative models. Examples include bar-level infilling (e.g., removing and rewriting one bar), melody generation conditioned on chords, variation generation from a given melody, counter-melody generation, or harmonizing a melody with chords. All of these tasks share the same music-conditioned generation pattern and may benefit from a strong generative pretrained model.

Beyond arrangement tasks, the proposed tokenization scheme may have broader implications for symbolic music modeling in general. By restructuring musical data into continuous track sequences, it facilitates learning over reusable musical phrases. This may improve both conditional and unconditional generation quality across tasks such as continuation, accompaniment generation, and reharmonization. Moreover, we anticipate that structured representations like REMI-z could benefit music understanding tasks, including symbolic transcription from audio, by facilitating the modeling of instrument-wise playing styles during decoding. Additionally, one may consider applying sequence compression techniques—such as Byte Pair Encoding (BPE) or variational autoencoders (VAE)—on top of this base tokenization to enhance compactness, which could further support long-range modeling essential for full-song generation. We encourage future work to explore these directions.

Beyond arrangement tasks, the proposed tokenization scheme may have broader implications for symbolic music modeling in general. By restructuring musical data into locally coherent, instrument-consistent sequences, it facilitates learning over meaningful musical units. This could improve both conditional and unconditional generation quality across tasks such as continuation, accompaniment generation, or reharmonization. Moreover, we anticipate that structured representations like REMI-z may benefit music understanding tasks, including symbolic transcription from audio, by facilitating modeling of instrument-wise playing styles when decoding. Also, you can consider combine sequence compression techniques, maybe BPE, maybe VAE, on top of that, to further enhance the compactness, which may facilitate long-range modeling that are important to full-song generation. We encourage future work to explore these directions.

However, there are potential risks. As with other generative models, the misuse of automatic arrangement systems could devalue human artistry if deployed without appropriate attribution or transparency. The system may also reflect and amplify stylistic biases present in the training data, which primarily consists of western popular music. This could marginalize underrepresented musical traditions or reinforce narrow definitions of musical aesthetics. We encourage future researchers and practitioners to consider ethical deployment strategies, such as transparent model labeling, dataset diversification, and collaborative workflows where AI augments rather than replaces human creativity.

# G Limitations

**Fine-tuning dataset focused on pop genres.** While our framework is designed to be genre-agnostic, our fine-tuning and evaluation primarily focus on pop-style arrangements. Future work can further explore the generalizability across a wider range of genres such as classical, jazz, or non-Western traditions.

**Strict structural alignment assumption.** Our framework assumes that the input and output music share identical musical structure, which enables a segment-to-segment reconstruction formulation. While effective, this simplification does not fully reflect the broader musicological concept of

arrangement, which often involves modifying the musical form or phrase structure. This constraint is not unique to our method—most prior works also rely on such structural alignment—but relaxing this assumption remains an important direction for future research.

**Lack of instrument planning.** Although our system allows users to specify the instruments and their voice relationships for each segment during inference, the global planning of these assignments is not addressed. In practice, achieving a musically satisfying arrangement often requires deliberate instrument planning across segments—e.g., rotating lead instruments, or changing instrument combinations between sections. This work focuses on controllability at the segment level and leaves high-level planning strategies for future exploration.

**Melody retention in piano reduction is not guaranteed.** While our piano reduction method captures harmonies and textures from ensemble music, it often fails to retain the melody. This is largely because the original piano track in multitrack data typically does not carry the main melodic line, and our current training setup does not include any explicit melody supervision. Solving this may require incorporating human-created piano arrangements or explicitly modeling melodic salience, which we leave for future work.

**Handling of Time Signature and Tempo.** Our current model does not explicitly model time signature or tempo tokens during fine-tuning. As a result, it is limited to handling music with a fixed 4/4 time signature and stable tempo. This design choice follows common practice in prior arrangement research [42, 35, 30], where similar assumptions are made to simplify modeling. Empirically, approximately 95% of the fine-tuning data is in 4/4 time (see Appendix A), and stable tempo is generally the norm in modern music [7]. Nevertheless, extending the framework to handle non-4/4 time signatures and dynamic tempo variations remains an important direction for future work.

