# OpenReview forum: "Unifying Symbolic Music Arrangement: Track-Aware Reconstruction and Structured Tokenization"
_NeurIPS.cc/2025/Conference — NeurIPS 2025 poster_

### Official Review · Reviewer_FStY · 2025-06-30

**Clarity:** 3
**Significance:** 3
**Originality:** 3
**Rating:** 4
**Confidence:** 2

**Summary:**

This paper introduces a unified framework for symbolic music orchestration, enabling flexible conversion from any input track to any target track via a “segment-level reconstruction” objective and a factorized “content–instrument–context” conditioning scheme. The key contributions are: (1) the REMI-z structured multi-track token representation, which maintains continuity within each instrument part and significantly reduces redundant tokens; (2) a lightweight fine-tuning approach on a pretrained symbolic music model that handles three representative orchestration tasks—band reinterpretation, piano simplification, and drum generation—without requiring parallel data or additional architectures; and (3) comprehensive evaluations on the Slakh2100 dataset showing that the proposed method outperforms task-specific state-of-the-art baselines in both objective metrics (Note F1, Instrument-IoU, Voice-WER, etc.) and subjective listening tests (fidelity, coherence, creativity, playability).

**Questions:**

1. How does REMI-z perform on segments with non-4/4 time signatures, mixed meters, or drastic tempo changes?

2. Can the framework support finer user controls, such as specifying dynamics, timbral blending, or ornamentation styles?

3. For full-song orchestration (beyond single segments), how can cross-segment consistency be maintained, and what designs might handle transitions between sections?

**Ethical Concerns:**

["NO or VERY MINOR ethics concerns only"]

**Final Justification:**

Thanks for the author's detailed response.
Based on the rebuttal, I will maintain the score

**Limitations:**

Reliance on symbolic MIDI input; it does not address errors or noise introduced by audio-to-MIDI conversion.

**Paper Formatting Concerns:**

no issues

**Quality:**

3

**Strengths And Weaknesses:**

Strengths: A single model, after self-supervised fine-tuning, supports multiple orchestration scenarios without parallel data or specialized modules.

Weaknesses: Fine-tuning only considers 4/4 time and ignores tempo and time-signature tokens, which may limit adaptability to unusual meters or tempo fluctuations.

---

> ### Author Rebuttal · Authors · 2025-07-28
>
> First of all, thank you for your review. We appreciate your recognition of our core contribution — both the tokenization design and the fine-tuning formulation — and your questions were truly insightful. Below, we respond to your comments in detail.
>
> --------------------------------------------
>
> **Weakness: Fine-tuning only considers 4/4 time and ignores tempo and time-signature tokens, which may limit adaptability to unusual meters or tempo fluctuations**
>
> You're absolutely right — our fine-tuning experiments only used 4/4 music, but there’s a clear rationale behind this. Many existing works on music arrangement [1,2,3] have also made the same assumption, and our choice follows this established convention. More practically, time signature follows an extremely long-tail distribution. As noted in Appendix A.1, about 95% of music in our fine-tuning dataset is in 4/4 — meaning only around 50 songs use a different time signature. This data size is probably insufficient for the model to learn to arrange for non-4/4 meters. Since our primary goal in this paper is to verify the viability of our proposed approach, we chose to focus on the most common scenario.
>
> You're also correct that our fine-tuning setup did not model tempo fluctuations — in fact, we did not include tempo tokens in fine-tuning. This also follows convention in prior works [1,2,3]. While tempo changes do occur in modern music, they’re relatively uncommon, and stable tempo is generally the norm [4]. So although being able to handle tempo fluctuations would certainly be a nice-to-have feature, it was not prioritized in this work.
>
> But it’s worth noting that our pretraining stage *does* use bar-level time signature and tempo tokens. So in principle, the model may have the potential to handle changes in time signature and tempo — if such information is also incorporated and with a larger scale fine-tuning. But since these factors are not central to the core problem of arrangement, we did not explore this direction in detail. We fully acknowledge this as a limitation and will include a discussion of it in Appendix G. This is certainly an open direction worth exploring in future work.
>
>
> **Question 1: How does REMI-z perform on segments with non-4/4 time signatures, mixed meters, or drastic tempo changes?**
>
> We haven’t tested our model on such cases. As mentioned earlier, the fine-tuning data used in our experiments was almost entirely in 4/4 and didn’t include tempo fluctuations. So while the model was pretrained with time signature and tempo tokens, we didn’t incorporate that information during fine-tuning.
>
> But it’s *likely* that REMI-z could handle such scenarios, especially with appropriate finetuning settings. Our time signature and tempo design follows MuseCoco [5], a model that has demonstrated the ability to handle varying time signatures and tempo changes (as shown on their demo page). So we’re optimistic that REMI-z might be capable of similar generalization, though confirming this would require further empirical validation — and we see that as an interesting direction for future work.
>
>
>
> **Question 2: Can the framework support finer user controls, such as specifying dynamics, timbral blending, or ornamentation styles?**
>
> In theory, yes — if these aspects can be extracted from the target music and included as part of the condition during self-supervised training, the framework should be able to support them. Although finer grained control may not be a priority in existing works [1,2,3], We did conduct some initial experiments on finer-grained control. Specifically, we explored conditioning on each instrument’s playing style (or “texture”) as part of a self-supervised setup. This allowed us to control the generation behavior of each track with greater nuance. The results were promising, but due to focus, time constraints, and space limitations, we did not include them in this paper. We believe these directions are valuable and worth exploring in future work, and would like to add an additional section in Appendix to discuss potential possibilities to add finer-grained control.
>
>
> **Question 3: For full-song orchestration (beyond single segments), how can cross-segment consistency be maintained, and what designs might handle transitions between sections?**
>
> This is a great question. Our observations differ depending on whether the arrangement involves pitched instruments (band arrangement and piano reduction) or drums.
>
> For pitched instruments, we found in early experiments that even without an explicit history condition, when given similar content sequences and instrument constraints, a trained model tends to generate stylistically similar arrangements. So when a musical segment reappears (e.g., a repeated chorus), the model tends to rearrange it in a consistent style. The long-term coherence is hence preserved just because the original composition has that coherence. We showcase this in several of our demos. By contrast, short-term cohesion between adjacent but dissimilar segments is a more pressing challenge, and this is precisely why we introduced the history condition. Without specific design choices, the model may apply entirely different arrangement “ideas” to adjacent segments, leading to a noticeable sense of discontinuity.
>
> For drum arrangement, since drum patterns are often more temporally dispersed, we used longer segments (4 bars). In this setting, the model appeared to learn how to detect and manage segment transitions on its own — a very interesting behavior. Increasing segment length may thus be another promising way to improve global coherence.
>
> We believe these observations are quite interesting, and we plan to include a new Appendix section titled “Discussion” to elaborate on them in the next version.
>
>
> **Limitation: Reliance on symbolic MIDI input; it does not address errors or noise introduced by audio-to-MIDI conversion**
>
> Thank you for pointing this out — we fully agree. Our work focuses on symbolic-level music arrangement and assumes clean MIDI input. Handling errors from upstream audio-to-MIDI conversion is beyond our current scope but is indeed an important practical consideration. We will add a note in the Limitation section to acknowledge this and encourage future research to explore multitrack arrangement in the audio-to-MIDI setting.
>
>
>
> --------------------------------------------
>
> That concludes our response — thank you again for the thoughtful and constructive comments!
>
> To summarize, we acknowledge that many finer aspects — such as time signature, tempo, and velocity — though not the center of the arrangement problem, were not addressed in the current version of the paper, but they are indeed meaningful and worth exploring. We’d be happy to include a more thorough discussion of these limitations in the camera-ready version (if given the opportunity).
>
> If you feel the clarifications helped address your concerns, we’d be truly grateful if you’d consider a slightly more positive rating. It would mean a lot to us. Thank you again for your time and support!
>
>
>
> [1] Zhao, J., Xia, G., Wang, Z. and Wang, Y., 2024. Structured multi-track accompaniment arrangement via style prior modelling. Advances in Neural Information Processing Systems, 37, pp.102197-102228.
> [2] Wang, Z., Xu, D., Xia, G. and Shan, Y., 2022, May. Audio-to-symbolic arrangement via cross-modal music representation learning. In ICASSP 2022-2022 IEEE International Conference on Acoustics, Speech and Signal Processing (ICASSP) (pp. 181-185). IEEE.
> [3] Terao, M., Nakamura, E. and Yoshii, K., 2023, June. Neural band-to-piano score arrangement with stepless difficulty control. In ICASSP 2023-2023 IEEE International Conference on Acoustics, Speech and Signal Processing (ICASSP) (pp. 1-5). IEEE.
> [4] Dannenberg, R.B. and Mohan, S., 2011. Characterizing tempo change in musical performances. In ICMC.

---

> > ### Comment · Reviewer_FStY · 2025-08-07
> >
> > Thanks for the author's responses

---

> ### Author Response · Authors · 2025-08-06
> **Follow-up on Your Review**
>
> Dear Reviewer,
>
> Thank you again for your thoughtful review. We wanted to kindly remind you that we have provided detailed responses to your concerns in the rebuttal. As the rebuttal phase is ending soon, we would greatly appreciate it if you could take a moment to review our responses and share any follow-up comments.
>
> Thank you for your time and support.

---

> ### Comment · Area_Chair_yYFL · 2025-08-06
>
> Dear Reviewer,
>
> This is a gentle reminder that the Author-Reviewer Discussion period will end on August 8, 11:59pm AoE.
>
> If you have not yet responded to the author’s rebuttal or comments, please take a moment to participate in the discussion. Simply submitting a “Mandatory Acknowledgement” without engaging in any discussion is not sufficient.
>
> Your active participation is important to ensure a fair and transparent review process.
>
> Thank you again for your valuable service.
>
> Best regards,
> AC

---

### Official Review · Reviewer_SnRL · 2025-07-03

**Clarity:** 3
**Significance:** 3
**Originality:** 3
**Rating:** 5
**Confidence:** 4

**Summary:**

"Unifying Symbolic Music Arrangement: Track-Aware Reconstruction and Structured Tokenization" presents a new method for symbolic music tokenization, building on ideas from REMI+. Models trained using REMI-z are compared to existing baselines on 3 musical tasks, across a wide suite of qualitative and quantitative evaluations. Across nearly all task metrics REMI-z shows significant gains, and musical examples also show the efficacy of this tokenization scheme.

**Questions:**

One key piece to proposing new tokenizations, can be showing their application at a variety of scales in more than one baseline model type - here it seems one, relatively small but high performing model architecture is all that was trained along with the complementary "no PT" model for each task. Are there any additional ablation studies at other model scales? Or integration into another existing model, to show it also generalizes well to this new tokenization? Transformers are very general, but given one of the comparison models is a UNet, can this tokenization also be applied to UNets and other non-Transformer models?

I did not see any direct generation results in the samples page, only other applications like reductions and arrangement. Do the authors have any generative samples to show, to get some idea of how this tokenization works for pure generative applications?

Bar encoding and adding in bar/timing marks can be a difficult endeavor - did the authors do anything special on certain datasets for cleanup and addition of these features? Or did all datasets have sufficient bar marking and note timing / quantization to apply REMI-z without additional preprocessing?

There is some comparison to TransformerVAE, but have the authors done any comparison to a learned encoder and quantization (similar to many common methods in multimodal modeling) using a frozen model with quantized output as a fixed tokenizer, directly in the same setting as the model training? This is mostly a curiosity question about learned structure, versus human specified.

Did the authors investigate instruction style finetuning at all (besides drum arrangement, which is kind of instruction-like)? Finding a melody and harmony to match a drum backing, laying a new melody over an existing harmonic structure, or harmonizing existing melody lines would be some other examples of useful "instruction tuning" for music tasks.

Particularly it seems there are significantly more tasks than raters in the rating pool - do the raters with musical expertise have a fundamentally different scoring than amateur / non-musical folks? Did every rater rate every single example, or what was the average amount of examples rated per rater?

On this line "... indicates reduced information redundancy, and suggests that REMI-z sequences consist of more predictable tokens, potentially facilitating the learning of generative models". Couldn't this also mean *more* redundant representations (for example, repeated tokens that may be merged in other tokenizers like BPE), rather than less redundant. There is often a fundamental tradeoff of compression/redundancy versus predictability, and it may be that less predicatable representations facilitate harder learning tasks, and thus better models in terms of robustness e.g. dropout, track deletion, and so on, so more specific measurement of this point and qualitative analysis would help if this is a critical and key point the authors wish to make.

**Ethical Concerns:**

["NO or VERY MINOR ethics concerns only"]

**Final Justification:**

Though I take the author's point that the contribution is not solely around tokenization, the title itself ends with "Structured Tokenization" and there are only limited model ablations directly (in terms of model scales / design variations). However, they have shows extensive ablations across tasks, and the writing is overall quite clear and fundamentally I like the paper. I have raised my score 1 point around this issue, as I do think this paper is something of interest in the symbolic music generation community - and will communicate what I think could make this work stronger still based on the first round of rebuttals.

**Limitations:**

Yes

**Paper Formatting Concerns:**

No major formatting concerns.

**Quality:**

3

**Strengths And Weaknesses:**

Strengths:
The writing is both detailed and clear, with strong organization. The method is straightforward and pragmatic, with thorough appendix content. I will write much more on weaknesses than strengths, but that is because the paper and approach are generally well-rounded! It is hard to comment on the strengths, because overall this paper is good, lacking only in a few small-but-critical to improve areas.

Weaknesses:
There are many, many metrics in the paper, but it is difficult to determine which are key. I lean toward direct human ratings first, for many of these tasks but human and quantitative metrics are mostly interleaved throughout the paper. There are human ratings across 4 separate sub-areas, where the model is ahead by a statistically significant amount in all categories - but are the overall ratings also positive? For example, there is a dimension for musicality but this may not align with "enjoyment" of the piece - some discussion of this would be useful. I would expect this "global score" to also be positive, but means a simple A/B head to head would be of interest as well. For the ratings, some description of the expected average musical ability of the raters (beyond just the music industry mention - this is not specific enough with regards to musical ability) used for the ratings would be useful to contextualize rating numbers.

Additional baselines would be useful - TransformerVAE and the UNet comparison are pretty limited in terms of comparisons, especially given the clear losses to the proposed method. Seeking out stronger baselines and more points of comparisons, given the broad flexibility of this tokenization and model, would help me boost my score - as the limited external points of comparison, along with only having 2 task models (w and w/o PT) limits the impact of the many metrics and human ratings performed.

Figure 1 is a bit hard to read, and going directly into tasks right away doesn't draw a reader in. It might be better to consider the tokenization figure (figure 3) as a better introduction, or even figure 5, as that is what the technical contribution of the paper is about - the tasks are a test of the technical contribution and model. Alternatively, some diagram showing the training and finetuning stages, the lifecycle of the model training, leading into Task 1, and so on, is also worth consideration.

Given the use of track-level deletion (which seems important, and likely important enough to move out of the appendix), some discussion and citation of related work like "Efficient Training of Language Models to Fill in the Middle (FIM)", BERT, Masked Autoencoder (MAE), discrete diffusion, and many other span-based methods in the background section could be worthwhile. Showing the *failure* when not doing this, would also be an interesting qualitative study.

---

> ### Author Rebuttal · Authors · 2025-07-29
>
> Thank you for your thoughtful and constructive feedback! We sincerely appreciate your kind words on our tokenization design. Your comment that “the paper and approach are generally well-rounded” truly made our day. Below we address your concerns in detail.
>
> ---
>
> **Clarifying the paper’s scope: The review’s “Summary” appears to primarily frame the contribution around tokenization.**
>
> While REMI-z is indeed an important component, we would like to clarify that the paper is centered on **symbolic music arrangement**, as reflected in the title and the Introduction section.
>
> Our key contributions are:
> (1) a unified model design across diverse arrangement tasks;
> (2) a self-supervised fine-tuning objective for arrangement (Sec. 3.1);
> (3) a track-aware tokenization scheme (REMI-z) that facilitates modeling (Sec. 3.2);
> (4) strong performance across three representative and challenging arrangement tasks.
>
> It’s possible that we did not sufficiently highlight the second contribution — the arrangement-specific fine-tuning objective — which is technically substantial and complementary to the tokenization. We’ll make sure to emphasize this more clearly in the Introduction of the next revision.
>
> We hope this clarification helps better contextualize our responses below.
>
>
>
> **Weakness 1: Relative importance of metrics, and the overall rating are unclear**
>
> We agree the rationale behind metric choices could be more explicit. For example, `Note_i F1` is the most critical, as it directly reflects closeness to ground-truth compositions [17,18]. We’ll clarify the metrics’ priorities and their motivations in Appendix C.
>
> As for the “overall rating,” the `Musicality` metric plays this role since the prompt asked “What is the overall musical quality?” (L661). In all three tasks, our model achieved the highest `Musicality` and was significantly preferred over major baselines.
>
> We agree A/B testing is valuable, but 5-point rating protocols are widely used in prior arrangement works [39,32,35]. Moreover, our results show strong significance (most p < 0.001), indicating clear improvement. We’ll include a new Section D.3 in the appendix to discuss the benefit of A/B testing and suggest it as future work.
>
> Lastly, thanks for flagging the vague “music industry” label. In the final version, we’ll clarify rater backgrounds (e.g., average years of training) to better contextualize the subjective results.
>
>
> **Weakness 2: More (stronger) external baselines would be useful**
>
> Thanks for the suggestion — though this may stem from a misunderstanding of our paper’s main contribution. Our focus is symbolic *music arrangement*, and the baselines we compare with are (to our knowledge) the strongest, task-specific SOTA methods: TransformerVAE [39] for band, UNet [26] for piano, and ComposersAssistant 2 [17] for drum. We believe this baseline selection is appropriate.
>
> If the paper were about proposing a general-purpose tokenizer, then broader task selections would be needed. But REMI-z is a supporting tool to enhance arrangement, but not the paper’s focus. Our goal is a unified, practical solution for diverse arrangement tasks.
>
> Each of the three tasks includes 4–5 arrangement candidates in human evaluation. The numbers of candidates are comparable to prior works like TransformerVAE [39], hence we believe the human evaluation is reliable.
>
> Since you noted this concern influenced your score, we hope this clarification helps. Our baseline choices were carefully matched to the scope and contribution of this work — advancing music arrangement via a unified model. We don’t claim generalization beyond that (e.g., to other model architectures or tasks). We hope this clarification resolves the concern — and we appreciate your consideration of our intent and scope.
>
>
> **Weakness 3: Figure 1 is a bit hard to read, consider replacing it.**
>
> We agree that a lifecycle diagram showing pretraining, fine-tuning, and inference would help clarify the overall structure. We’ll add one before Figure 1 in the revised version to guide the reader through the stages of our method.
>
> But we believe Figure 1 still serves a key purpose and should remain in place. It conveys the high-level unification that the same model and training strategy can address multiple arrangement tasks, which is the central message of the paper.
>
>
> **Weakness 4: Track-level deletion seems important, and discussion of span-based methods may be needed.**
>
> The reason we placed this in the appendix is that track deletion is only used in the *band arrangement* task. We’re happy to add discussions of other span-based methods like BERT to the Related Work (or Appendix B.2 if cannot meet the space constraint) in the next version.
>
>
> **Q1: Additional test of REMI-z at different model scales or in alternative architectures**
>
> **On evaluation scope:**
> This paper is not a tokenization paper, but an application-driven work focused on improving symbolic music arrangement. Although we showed REMI-z has potential beyond arrangement tasks (Sec. 5.4), further exploration of it is beyond the scope of the paper.
>
> **On model scaling:**
> Since our focus is on method, not scale, we deliberately use small models and modest-scale pretraining to show that improvements stem from the approach itself. Larger models generally yield better results, but replicating that trend may not strengthen our core contribution.
>
> **On what was trained :**
> We conducted pretraining, fine-tuning on 3 tasks (with/without pretraining), re-trained TransformerVAE and UNet, and additionally trained 1 conditional + 2 unconditional models for tokenization comparison. Other baselines were used as-is, with rationale in Appendix D.4. While some other works may report hundreds of experiments, our aim is to show that this unified setup outperforms strong task-specific systems — which we believe our experiments have sufficiently demonstrated.
>
>
>
>
> **Q2: Do the authors have “direct generation results”? Any special handling for bar/timing marks?**
>
> All results on our demo page are *direct generation outputs* from our models and baselines, across all *arrangement tasks* mentioned in the paper.
>
> For bar and timing encoding (Section 3.2, Appendix A): Bars are delimited by `End-of-Bar` tokens. Note onsets are encoded as relative positions within a bar using 48-th note quantization (supporting 16th-note triplets, same as REMI+). Durations are encoded similarly. This scheme applies uniformly across all datasets, without task-specific cleanup.
>
>
> **Q3: Have the authors tested learned quantization/tokenizer approaches?**
>
> We haven’t tried with frozen encoders or codec models in this work.
>
> We view REMI-z and learning-based quantization/tokenization as orthogonal (as L111). REMI-z is a symbolic-level base tokenization — it defines musical events with a token sequence. Learned quantization/tokenization (VQ-VAE, codec, or BPE) typically operates *on top of* such base sequences. But our focus is on improving base token structure to improve arrangement modeling. Combining REMI-z with learned quantizers/tokenization is possible, but outside this paper’s scope. We will clearly explain this in Sec. 2.2 in the next version.
>
>
> **Q4: Instruction finetuning, more tasks, and subjective evaluation details**
>
> **Instruction finetuning.**
> We agree this is a promising direction. We didn’t try it in this work because different tasks involve different segment lengths, which makes batching inefficient.
>
> **Applicability to more tasks.**
> Our focus is on *arrangement*, specifically generating musically meaningful versions of a piece under new instrumentation. The three chosen tasks cover diverse and representative scenarios. While other generation tasks may be potentially supported, we chose to stay focused for clarity. We see general music-to-music generation as a promising extension.
>
> **Musical background of raters**
> 73.1% of raters reported >10 years of musical training (L685). We ran t-tests comparing these raters to those with ≤10 years (using piano reduction ratings on our model), and found no significant differences of the mean of the two clusters (*p* > 0.2 across all metrics). This suggests musical background didn’t systematically bias results.
>
> Each rater evaluated a subset of groups, where one group = one song (input) with multiple model outputs. On average, raters evaluated 2.15 groups for band, 2.80 for piano, and 2.96 for drum — corresponding to 8.6, 14.0, and 11.8 samples respectively (L683). We will add these details in Appendix D.2.
>
>
> **Q5: Each REMI-z token seems to carry more information and hence should be more redundant?**
>
> We acknowledge that “redundancy” here was not precisely used. REMI-z has lower redundancy mainly due to its shorter sequence length, achieved without introducing new token types compared to REMI+. Meanwhile, its lower Shannon entropy indicates a more predictable token distribution. As a result, REMI-z uses fewer and more predictable tokens, making it potentially easier for generative models to learn — and validated by its lower note-level entropy. As noted in Q3, REMI-z is a base tokenizer and can be combined with BPE for further improvements. Since tokenization is not the paper’s focus, we stopped exploration at this point. We will revise Section 5.4 for clarity.
>
>
> ---
>
> That concludes our response — thank you again for the thoughtful and constructive comments!
>
> To summarize, the focus of the paper is about a unified method for symbolic music arrangement, rather than a general-purpose tokenization scheme, and we believe this adequately justify our experiment designs. We’d be more than happy to revise the framing and emphasize the major contribution in the camera-ready version (if we’re fortunate enough to have the opportunity).
>
> If you feel the clarifications helped address your concerns, we’d be truly grateful if you’d consider a slightly more positive rating. It would mean a lot to us. Thank you again for your time and support!

---

> ### Author Response · Authors · 2025-08-06
> **Follow-up on Your Review**
>
> Dear Reviewer,
>
> Thank you again for your thoughtful review. We wanted to kindly remind you that we have provided detailed responses to your concerns in the rebuttal. As the rebuttal phase is ending soon, we would greatly appreciate it if you could take a moment to review our responses and share any follow-up comments.
>
> Thank you for your time and support.

---

> ### Comment · Area_Chair_yYFL · 2025-08-06
>
> Dear Reviewer,
>
> This is a gentle reminder that the Author-Reviewer Discussion period will end on August 8, 11:59pm AoE.
>
> If you have not yet responded to the author’s rebuttal or comments, please take a moment to participate in the discussion. Simply submitting a “Mandatory Acknowledgement” without engaging in any discussion is not sufficient.
>
> Your active participation is important to ensure a fair and transparent review process.
>
> Thank you again for your valuable service.
>
> Best regards,
> AC

---

> ### Comment · Reviewer_SnRL · 2025-08-06
> **Reply to Authors**
>
> Thank you to the authors for the detailed reply, this cleared up a number of questions and helped me re-focus the discussion!
>
> Given the discussion here, it is worth considering a tweak to the title, or more explicitly calling out that tokenization is only an indirect focus of the paper and the self-supervised + model combination is really the more core focus. I do take the point that tokenization ablations in the detail I mentioned above, are not in scope for the paper but tokenization is also directly mentioned in the title AND second bullet of the intro as a core contribution, and from what I can see some of the performance is directly enabled by the novel tokenization strategy here given the REMI / REMI-z ablations run.
>
> On 1) I think it is somewhat difficult to describe a decoder only transformer as a "unified model design" contribution, as we have seen this trend for many years (going back to enc/dec LSTM and seq2seq work, pre Transformer even). A "unified and simplified baseline modeling approach" is probably a better description for the adoption of a decoder-only transformer, trained using a particular combination of losses and data ordering to outperform existing approaches.
>
> Leading into below discussion: Generally, I would also like to see some amount of model scale variations if modeling is a focus, especially given the small size of the chosen model - how was 80M chosen here? Number of layers? Do you see benefits or degradations from scale? Could it be even smaller? Even if "scaling laws" are not a core focus, some amount of ablation for the model size chosen is useful for broader adoption, and for understanding how research into these particular tasks may evolve in future work.
>
> I take the point that "hundreds of models" is not the goal here, and the ablations around conditional and unconditional models is a good start. Re-training the baselines (TransformerVAE and UNet) is useful but actually in my mind less necessary than ablations on the chosen model size and architecture itself, as long as the baseline papers also trained on the same dataset - since the numbers the previous papers got are clearly their best result, as long as the comparison is roughly equal then any retraining is more about reproducibility of work, rather than numerical comparison of best runs. Ablations around model size / depth particularly in Transformers have numerous strong knock on consequences for followup work (including potentially gathering much larger datasets if scaling does potentially show merit), and thus is influential on researchers reading the paper planning future work, and one of the reasons we do see 100s of training runs in certain areas of Transformer modeling.
>
> Given the short turnaround of the discussion phase I am not at all expecting modeling answers during this rebuttal, but these things are definitely worth thinking about in general, and could enhance the final demonstrations of this work.
>
> On A/B testing - there are studies in other areas which show that effects seen in opinion to opinion comparisons (like the 1 to 5 here) may show the opposite or non-significant results when done in a head to head A/B, which is what lead to my question. Similar discussions exist around "scaled" A/B (which allow strong, weak, or even neutral preference between items) vs binary choice. Mostly I see the two tests as complementary but revealing different information, rather than redundant - so having A/B comparison alongside opinion scores can be useful. There are several papers on this, but some acoustics related work in speech can be found at "Experimental evaluation of MOS, AB and BWS listening test designs", Wells et. al.
>
> On Figure 1: I agree it absolutely has its place, but as a "lead in" to draw in readers and researchers, it really needs the context from reading the paper to fully understand. Having a lighter weight "overview" for figure 1 as mentioned in your rebuttal should really help the paper flow, though it was already pretty good on this aspect!
>
> As for Q3, the reasons brought up for comparison is that VQ or BPE based methods often *replace* domain specific tokenizations (rather than strictly being placed on top), though sometimes incorporating limited aspects of the domain (like time-frequency transforms in general audio tokenization and codecs). Showing that this generalized semi-supervised objective and unified framing isn't critically dependent on being used with REMI-* tokenization variants - thus fully splitting the model + semi-supervised contribution from the tokenization, could enhance the adoption of this framework generally. This is what lead to this line of questioning.

---

> ### Comment · Reviewer_SnRL · 2025-08-06
> **Reply to Authors (part 2)**
>
> Demonstrating more details about the tokenization may interest readers more broadly in adopting the method (as tokenizers are often much easier to adopt than other things), so any more detailed study only enhances the work. My particular interest was around use of REMI-z as a modified format for pure generation, which lead to my general line of questioning, and the Q2 here. Though I take the point that these are "generated rearrangements" I am interested in models which generate the initial notes+durations+metadata for performance as well, beyond the drum discussion in section 5 and table 5, similar to Pop Music Transformer where REMI originated.
>
> On Q5 having lower entropy and better compression rates is not always a benefit - for example, see broad discussions about the "likelihood trap", to comparisons of rate vs codebook coverage in VQ methods. Given the focus of the paper is not tokenization I won't dwell on it, but fewer and more predictable tokens can sometimes harm generation and generalization, depending on the broader context. This may for example influence the choice of REMI vs REMI-z, if REMI-z is prefereable only for arrangement or in general modeling as well. The answers are rarely cut and dry when analyzing tokenizers, since they often critically interact with models trained ON said tokens, and we ultimately are analyzing the token + model combination when adopting any new ideas.

---

> > ### Author Response · Authors · 2025-08-08
> >
> > Thank you for the detailed reply, for recognizing that the main focus of our paper is on arrangement tasks, and for accepting our point that running as many experiments as possible is not the goal. We believe your constructive suggestions will be very helpful for future work.
> >
> > Regarding tokenization, we appreciate your suggestion. While the title (currently in the *topic: method 1 and method 2* format) may not be easy to change, we will revise the Introduction and Abstract to explicitly clarify that tokenization is a secondary focus, in order to avoid misunderstanding. We also acknowledge that our current “model + self-supervised objective” has only been tested on REMI-* tokenizations, so its performance under alternative tokenizations remains unknown. We will explicitly note this in the Limitation section. Exploring this decoupling, as well as further studying REMI-z—particularly its perceptual impact in unconditional generation with more human-rater evaluations—is part of our planned future work. The “likelihood trap” phenomenon you mentioned from NLP is a valuable reminder, and we will take it into account in these follow-up explorations.
> >
> > On modeling, we agree that a “decoder-only Transformer” is not a novel architecture (which is why contribution (1) in the rebuttal is not listed in the paper’s Introduction) and that “unified and simplified baseline modeling approach” is indeed a better description. You noted that “even if scaling is not a core focus of the paper, the choice of the 80M model should be explained.” While we mention the configuration in Section 4.1 and Appendix B.5, we did not explain the choice in detail. Our setup follows GPT-1’s 12-layer, 768-dim embedding, 3072-dim inner states  [1], and our experimental design is also in the same style: a single pretrained model and multiple fine-tuning tasks to show the efficacy of the pretrain–finetune paradigm. We used a slightly higher number of attention heads (16 instead of 12) based on the intuition that the interactions between different music tokens are more critical for musical quality than the value of individual note attributes. We will add these explanations to the appendix. We agree that more ablations on model size would be valuable for building competitive systems and informing production use, and we will include such experiments in future work.
> >
> > On A/B testing, we agree with your perspective. As noted in the rebuttal, we will include in the appendix a discussion of its complementary relationship with MOS, and we will consider adding A/B tests in future experiments.
> >
> > Once again, thank you for your valuable comments. We believe these insights help us improve the current work, and also provide important directions for subsequent research.
> >
> > [1] Radford, A., Narasimhan, K., Salimans, T. and Sutskever, I., 2018. Improving language understanding by generative pre-training.

---

### Official Review · Reviewer_Xe8N · 2025-07-05

**Clarity:** 3
**Significance:** 3
**Originality:** 3
**Rating:** 4
**Confidence:** 3

**Summary:**

This paper proposes a method for automatic multitrack music arrangement. The authors aim to integrate different aspects of this task—band arrangement, piano reduction, and drum arrangement—into a single unified model. Recognizing the limitations of strictly time-ordered tokenization schemes such as REMI+, the authors introduce **REMI-z**, a structured tokenization approach that organizes musical notes according to pitch and instrument. They also propose a unified fine-tuning method on the REMI-z sequences. Experimental results demonstrate that this method outperforms existing approaches across all three tasks, highlighting the effectiveness of both the REMI-z encoding and the fine-tuning strategy.

In summary, this work presents two key contributions to multitrack music arrangement: a novel encoding scheme and a unified fine-tuning method.

**Questions:**

See weaknesses and suggestions.

**Ethical Concerns:**

["NO or VERY MINOR ethics concerns only"]

**Limitations:**

yes

**Quality:**

3

**Strengths And Weaknesses:**

Strengths

1. The paper presents a valuable motivation by proposing a unified model for diverse music arrangement tasks.

2. The authors provide clear descriptions of dataset construction, encoding examples, and relevant implementation details, ensuring full reproducibility.

3. The paper maintains strong alignment between its methodology and experiments. All three tasks, the fine-tuning components, and the new encoding scheme are carefully evaluated, with consistent and convincing improvements shown.

Weaknesses

1. **Unclear Method Description**: The definition of the task-specific source sequence is unclear when it is first introduced. Moreover, it remains ambiguous how the proposed method could be generalized to other music-related tasks beyond the three examined here. (See suggestions below for more detail.)

2. **Evaluation Concerns**: The three evaluation tasks appear to be newly defined by the authors without references to prior work or benchmarks. In particular, for the *Band Arrangement* task, if the model is provided with $y$ as $S_{\text{task}}(y)$, the task may become trivial—even with randomness introduced during training—since the model can directly refer to the source and assign corresponding instruments. This could explain why it outperforms methods like VAE on note-F1, which encode source information into latent vectors. Therefore, I remain skeptical about whether it is reasonable to include target note information directly in the input during evaluation.

Suggestions

1. On line 48, where you state *“We further observe that strictly time-ordered tokenizations introduce redundancy and fragment track content,”* I suggest adding supporting evidence—either references or quantitative/visualized examples. Readers may not be fully convinced why REMI+ is unsuitable here; simply stating that it is “unstructured” is insufficient, since REMI+ is time-structured by design. This clarification is especially crucial because it supports your motivation.

2. Please provide a clearer explanation of the task-specific source and target tracks around lines 145–146. From Figure 2, it appears that *CONTENT* is merely a rearrangement of the *TARGET SEQUENCE* without instrument labels, making it confusing why the answer is embedded in the input. Only in Section 3.3 does it become clearer that the source can take other forms. It would greatly improve readability if you clearly defined this source-target relationship at the first mention—especially since you aim to build a unified framework. You might also consider reorganizing Sections 3.1 and 3.2; presenting the training method before the tasks could improve the logical flow, as the training process directly depends on task formulations, while REMI-z could naturally follow afterward as the core technical detail.

---

> ### Author Rebuttal · Authors · 2025-07-28
>
> First of all, thank you for your comments. We appreciate your recognition of our core contribution, and your suggestions are very constructive. Below, we address the specific concerns you raised.
>
> --------------------------------------------
>
> **Weakness 1: The definition of the task-specific source sequence is unclear at first introduction, and how can the method generalize to other tasks**
>
> You're absolutely right. Although the formula for the task-specific source sequence appears in Section 3.1, we don’t actually explain what $S_{\text{task}}$ and $T_{\text{task}}$ mean until Section 3.3, where we introduce the individual tasks. In the next version, we’ll briefly clarify right after Eq (1) what these two filters are and why they’re necessary — specifically, that not all tasks require all tracks as conditions or outputs, and that the exact sets depend on the needs of each task.
>
> As for generalization, we’d like to clarify that the central focus of this paper is still **arrangement** — that is, creating music variations for different instrument settings — and we selected three representative tasks to cover different scenarios in this domain. Generalizing to other music-to-music generation tasks is definitely part of our long-term goal. Many such tasks can be formulated similarly: we construct a conditional sequence (often derived from a partial or aspect-wise decomposition of the target), concatenate it with the remaining target tokens, and then apply segment-level generation using a pretrained model with context-awareness from the history. REMI-z may also contribute to performance improvements in these broader scenarios.
>
> Examples of such tasks include bar-level infilling (e.g., removing and rewriting one bar), melody generation conditioned on chords, variation generation from a given melody, counter-melody generation, or harmonizing a melody with chords. These all share the same music-conditioned generation pattern. Due to time constraints, we only conducted experiments and evaluations on the three most representative arrangement cases. In the current version, we only hint at broader applicability in the abstract and conclusion using intentionally soft phrasing ("suggest", "potential"). Actually, we have also explored finer-grained control in a separate setting, where each instrument’s texture (i.e., playing style) is treated as a condition during self-supervised training to control generation behavior per track. The results were encouraging, but we chose not to include them here due to limitations in focus, space, and time. In the next version, we will discuss the potential generalizability of our method in depth in Appendix F to avoid potential confusion.
>
>
> **Weakness 2: Evaluation tasks are newly defined and may be trivial, especially for band arrangement where target note information is seemingly present in the input**
>
> Thank you for raising this concern — we acknowledge that our current writing didn’t cite the task origins explicitly. In fact, the three tasks we use are based on prior work: band arrangement follows the setup in [1,2] with a more challenging setting (without track-wise prior as prompt), piano reduction is adapted from [3,4], and drum generation serves as a representative case of track-level infilling, inspired by [5,6]. We’ll make sure to include these references in the task descriptions in the next version.
>
> Regarding the objective evaluation setup for band arrangement, we believe the current input format is proper and follows the previous works [1,2], where it is common to provide note event information without instrument labels, and have the model infer an instrument-specific arrangement from this content. In our notation, the input is not $S(y)$, but rather $C(S(y))$. The $C()$ function removes all instrument identity, imposes a strictly time-ordered view (so that note ordering doesn’t implicitly encode track information as well), removes duplicate events, and even removes note duration. As shown in Figure 5 of Appendix A, applying $C()$ significantly alters the appearance of the target sequence — making the task non-trivial.
>
> Recovering $y$ from $C(y)$ is far from easy. In music theory, this task is often referred to as instrumentation or orchestration [7]. It requires extensive knowledge of instrument properties, arrangement principles, and often involves collaboration with musicians who are familiar with specific instruments. For a model, random allocation would clearly fail, and naive solutions like rule-based heuristics (Table 2) don’t perform well. A good model must learn not only each instrument’s physical constraints (e.g., playable pitch range), but also their typical role in an ensemble — whether melodic, harmonic padding, flourish, or rhythmic base — in order to produce high-quality arrangements. This objective setup implicitly evaluates whether the model understands these roles from training.
>
> While objective metrics may not perfectly correlate with perceived musical quality (that’s why we also conducted human evaluations to assess creativity and musicality), they are still meaningful. We believe the objective evaluation results here suggest that our model successfully captures instrument-role relationships with minimal external guidance — unlike [1], which requires note density and pitch histogram prompts, or [2], which relies on fixed instrument combinations throughout the training set.
>
> Your point is an excellent reminder that this setup may not be self-evident to readers unfamiliar with music theory. We will cite the papers related to the task settings, add a short paragraph in Section 3.3 to clarify the difficulty of the band arrangement task, and move the fuller explanation above into a new Appendix section.
>
> As a minor clarification: there is no randomness in the objective evaluation for band arrangement (see L234); creativity only enters during human evaluation to avoid introducing noise in F1-based metrics.
>
>
> **Suggestion 1: The claim that time-ordered tokenizations introduce redundancy and fragment content lacks supporting evidence**
>
> Thanks for the suggestion. In the Introduction, we prioritized logical flow and therefore didn’t go into detail at that point. However, the actual issue of content fragmentation — what it means and why it matters — is explained later in Section 2.2 (starting from line 99), along with Figure 3 as an illustrative example.
>
> In the next version, we’ll explicitly refer readers to Section 2.2 when we first mention this issue, to help avoid confusion and ensure the motivation is clearly supported.
>
>
> **Suggestion 2: Clarify the source- and target-specific tracks and consider reorganizing Sections 3.1 and 3.2**
>
> Thank you for pointing out this potential source of confusion. This is indeed closely related to our response to Weakness 2. The “content sequence” is, as you observed, a kind of “rearrangement” by flattening all note events to a single time-ordered stream — and learning how to reconstruct the full original music from it is exactly what our model is designed to do. This provides a self-supervised formulation of the arrangement task, which, as mentioned earlier, is far from trivial. In music theory, this kind of reconstruction requires significant domain expertise, and existing models may struggle with it. We’ll make sure to emphasize the difficulty of this task more clearly in Section 3.3, and include a more detailed explanation in a new Appendix section (since the main text may not have enough room).
>
> Regarding the ordering of sections — we think your suggestion makes a lot of sense. While $S_{\text{task}}$ and $T_{\text{task}}$ are introduced formally in Section 3.3, they first appear in Section 3.1. In the next version, we will explain their meaning and necessity at first mention. We also plan to restructure Section 3.1 to present the training method and Equation (1) earlier, followed by the different task-specific conditioning schemes. We believe this will significantly improve readability and logical flow.
>
>
>
> --------------------------------------------
>
>
> That concludes our response — thank you again for the thoughtful and constructive comments!
>
> To summarize, we realize we may not have conveyed the full complexity of the band arrangement task as clearly as we could have, and we’d be happy to revise and expand on this in the camera-ready version (if given the opportunity).
>
> If you feel the clarifications helped address your concerns, we’d be truly grateful if you’d consider a slightly more positive rating. It would mean a lot to us. Thank you again for your time and support!
>
>
>
> [1] Zhao, J., Xia, G., Wang, Z. and Wang, Y., 2024. Structured multi-track accompaniment arrangement via style prior modelling. Advances in Neural Information Processing Systems, 37, pp.102197-102228.
> [2] Dong, H.W., Donahue, C., Berg-Kirkpatrick, T. and McAuley, J., 2021. Towards automatic instrumentation by learning to separate parts in symbolic multitrack music. arXiv preprint arXiv:2107.05916.
> [3] Terao, M., Nakamura, E. and Yoshii, K., 2023, June. Neural band-to-piano score arrangement with stepless difficulty control. In ICASSP 2023-2023 IEEE International Conference on Acoustics, Speech and Signal Processing (ICASSP) (pp. 1-5). IEEE.
> [4] Chiu, S.C., Shan, M.K. and Huang, J.L., 2009, December. Automatic system for the arrangement of piano reductions. In 2009 11th IEEE International Symposium on Multimedia (pp. 459-464). IEEE.
> [5] Malandro, M.E., 2023. Composer's Assistant: An Interactive Transformer for Multi-Track MIDI Infilling. arXiv preprint arXiv:2301.12525.
> [6] Malandro, M.E., 2024. Composer's Assistant 2: Interactive Multi-Track MIDI Infilling with Fine-Grained User Control. arXiv preprint arXiv:2407.14700.
> [7] Wikipedia. 2024. *Orchestration*. [online] Available at: <https://en.wikipedia.org/wiki/Orchestration> [Accessed 25 Jul. 2025].

---

> ### Author Response · Authors · 2025-08-06
> **Follow-up on Your Review**
>
> Dear Reviewer,
>
> Thank you again for your thoughtful review. We wanted to kindly remind you that we have provided detailed responses to your concerns in the rebuttal. As the rebuttal phase is ending soon, we would greatly appreciate it if you could take a moment to review our responses and share any follow-up comments.
>
> Thank you for your time and support.

---

> ### Comment · Area_Chair_yYFL · 2025-08-06
>
> Dear Reviewer,
>
> This is a gentle reminder that the Author-Reviewer Discussion period will end on August 8, 11:59pm AoE.
>
> If you have not yet responded to the author’s rebuttal or comments, please take a moment to participate in the discussion. Simply submitting a “Mandatory Acknowledgement” without engaging in any discussion is not sufficient.
>
> Your active participation is important to ensure a fair and transparent review process.
>
> Thank you again for your valuable service.
>
> Best regards,
> AC

---

### Comment · Area_Chair_yYFL · 2025-08-05
**Reminder: Author-Reviewer Discussion Ends Aug 6**

Dear Reviewers,

This is a quick reminder that the Author-Reviewer Discussion phase (July 31 – Aug 6) is ending soon. Feel free to raise any comments or questions, these can help prompt author responses.

Best, Area Chair

---

### Note · Authors · 2025-08-12

We appreciate the time and effort all reviewers have already invested in this process.

For Reviewer Xe8N, there was no engagement in the discussion phase. On our end, there was no record of a mandatory acknowledgement or a final justification. This is noted here for the AC’s consideration.

Reviewer SnRL initially may have some misunderstanding of the paper’s main contribution and raised several concerns regarding experiment design and baseline choices. We believe our rebuttal addressed these concerns, after which the reviewer provided constructive feedback and suggestions for future work, which we sincerely appreciate.

Reviewer FStY did not participate in the discussion beyond a brief thank‑you message. In the mandatory acknowledgement, they indicated that the final justification had been submitted; however, the score remained visible on our end, which—based on the review process—may indicate that the final justification was not completed. This is noted for the AC’s consideration.

Finally, we respectfully encourage all reviewers to revisit the rebuttal and discussion in light of the clarifications provided, and to ensure that their final assessments and justifications are recorded in the system so the AC has complete information during decision-making. Thank you for your time and consideration.

---

### Decision · Program_Chairs · 2025-09-17

**Decision:**

Accept (poster)

**Comment:**

This paper proposes a unified symbolic music framework with a new tokenization scheme (REMI-z) and segment-level reconstruction objective, enabling a single pre-trained model to perform diverse multitrack music arrangement tasks and outperform task-specific baselines.

**Comments:**

- Reviewer Xe8N (Final Rating: 4 - Borderline accept): The paper makes a valuable contribution by presenting a unified framework with strong methodology–experiment alignment and reproducibility, though clarifications on task definitions and evaluation design would strengthen it.

- Reviewer SnRL (Final Rating: 5 - Accept): The work is clearly written, well-organized, and demonstrates thorough ablations and strong results, with the main suggestions being to add stronger baselines and clarify human evaluation.

- Reviewer FStY (Final Rating: 4 - Borderline accept): The model convincingly supports multiple orchestration scenarios with a single framework, with the only limitation being its current focus on 4/4 time and lack of tempo/time-signature handling.


**Meta Comments:**

The reviewers agree that the paper is well-written, reproducible, and presents a solid and pragmatic unified framework for multitrack music arrangement, showing consistent improvements and strong potential impact in the symbolic music community. While concerns remain about limited baselines, unclear task definitions, scope of evaluation, and some missing aspects of musical representation (e.g., time signatures, tempo), the strengths outweigh the weaknesses.

Overall, the consensus leans positive, with ratings ranging from borderline accept to accept, leading to an overall decision of acceptance (Poster).